# MEDOE: A Multi-Expert Decoder and Output Ensemble Framework for Long-tailed Semantic Segmentation

## Abstract

Long-tailed distribution of semantic categories, which has been often ignored in conventional methods, causes unsatisfactory performance in semantic segmentation on tail categories. In this paper, we focus on the problem of long-tailed semantic segmentation. Although some long-tailed recognition methods (*e.g.*, re-sampling/re-weighting) have been proposed in other problems, they are likely to compromise crucial contextual information in semantic segmentation. Therefore, these methods are hardly adaptable to the problem of long-tailed semantic segmentation. To address this problem, we propose a novel method, named MEDOE, by ensembling and grouping contextual information. Specifically, our MEDOE is a two-sage framework comprising a multi-expert decoder (MED) and a multi-expert output ensemble (MOE). The MED includes several "experts", each of which takes as input the dataset masked according to the specific categories based on frequency distribution and generates contextual information self-adaptively for classification. The MOE then ensembles the experts' outputs with learnable decision weights. As a model-agnostic framework, MEDOE can be flexibly and efficiently coupled with various popular deep neural networks (*e.g.*, Deeplabv3+, OCRNet, and PSPNet) to improve the performance in long-tailed semantic segmentation. Experimental results show that the proposed framework outperforms the current methods on both Cityscapes and ADE20K datasets by up to 2% in mIoU and 6% in mAcc.

## 1   INTRODUCTION

Semantic segmentation is defined as the task to predict the semantic category for each pixel in an image. As a fundamental computer vision task, semantic segmentation is of great importance in various real-world applications (*e.g.*, clinical analysis and automatic driving). Conventional methods follow the *backbone-context module* architecture, where the context module contributes significantly and can be modified to enhance the extraction and aggregation of surrounding pixels and global information through large-scale field convolution model or attention mechanisms (Chen et al., 2018; He et al., 2019; Huang et al., 2019; Yuan et al., 2019; Zhao et al., 2017). Such modifications have enabled the recent state-of-the-art methods to improve the performance in semantic segmentation on various benchmark datasets (Caesar et al., 2018; Ding et al., 2019; Zhu et al., 2019).

Despite the overall impressive performance, the above-mentioned semantic segmentation methods still face challenges from the perspective of data distribution. For example, Table 1 shows the results for *head*, *body*, and *tail* categories (*i.e.*, categories ranked from top to bottom by pixel frequency) on benchmark datasets using DeepLabv3+ (He et al., 2016), where positive correlations can be found between results and data distribution. In other words, the performance declines for tail categories. This suggests the problem of *long-tailed distribution* in

Table 1: Comparison of results in terms of mIoU (%) and mAcc (%) for head, body, and tail categories on Cityscapes (Cordts et al., 2016) and ADE20K (Zhou et al., 2017) for semantic segmantation using Deeplabv3+ and ResNet-50c.

|  | Head | | Body | | Tail | |
|---|---|---|---|---|---|---|
|  | mIoU | mAcc | mIoU | mAcc | mIoU | mAcc |
| Cityscapes | 95.06 | 97.74 | 83.92 | 91.49 | 73.74 | 81.43 |
| ADE20K | 65.50 | 78.71 | 44.75 | 59.75 | 33.64 | 42.97 |

Figure 1: Example of semantic segmentation on Cityscapes, where tail categories (*e.g.*, "pole" and "wall") are not well segmented. From left to right: street scene image, ground truth, and segmentation map using DeepLabv3+.

semantic segmentation: a few *head* categories dominate the majority of pixels, whereas many *tail* categories correspond to significantly less pixels. Specifically, if all categories are processed in the same pattern while a long-tailed distribution exists, head categories may excessively influence the training and negatively impact on learning the contextual information about tail ones, leading to unsatisfactory pixel-level classification. Solving the problem of long-tailed semantic segmentation is critical to real-world scenarios. As shown in Figure 1, distinguishing poles from street scene images may prevent potential accidents.

To the best of our knowledge, this paper is the first to explicitly focus on the long-tailed semantic segmentation, and aims to provide a straightforward solution by extending the existent recognition methods, which adopt various strategies to solve the problem of long-tailed distribution in image classification (Buda et al., 2018; Huang et al., 2016; Gupta et al., 2019). These methods can be generally categorized into *re-weighting*, *re-sampling*, and *ensembling and grouping*. The re-weighting methods increase the weights of tail categories but decrease those of head ones (Cao et al., 2019b; Liu et al., 2019), following the assumption that images are nearly independent and identically (*i.e.*, i.i.d.) distributed to address the imbalance of training set. This assumption enables the classification accuracy for each category to depend on the frequency of the corresponding images (Cao et al., 2019b; Cui et al., 2022). The re-sampling methods conduct under-sampling for the head categories and over-sampling or even data augmentation for the tail ones (He & Garcia, 2009; Kim et al., 2020; Chu et al., 2020), usually following a random sampling strategy to ensure fairness. The ensembling and grouping methods start with training a feature extractor on the whole of an imbalanced dataset as representation learning, then adjust the margins of classifiers using multi-expert frameworks (Xiang et al., 2020; Zhang et al., 2021; Zhou et al., 2020) for re-balancing (Kang et al., 2019).

Nevertheless, the afore-mentioned recognition methods for long-tailed image classification can hardly be adapted for long-tailed semantic segmentation. Specifically, the re-weighting methods are not able to serve pixel-level classification as a pixel is usually highly correlated to the surrounding ones (*i.e.*, not i.i.d.) given the contextual information in the image (Lin et al., 2017). They may also cause a *see-saw* phenomenon: the accuracy for head categories is compromised whereas tail categories are on purpose emphasized. Based on random sampling, the re-sampling methods lead to a large number of independent pixels that undermine the contextual information of an image and can be detrimental to semantic segmentation. The ensembling and grouping methods rely heavily on re-balancing, where classifier re-adjusting is not well adapted for semantic segmentation due to ignoring the difference among head, body, and tail categories in contextual information.

Motivated by the observation about the ensembling and grouping methods, we propose MEDOE, a two-stage multi-expert decoder and output ensemble framework for long-tailed semantic segmentation. At Stage 1, the feature map extracted by a backbone trained on the whole of an imbalanced dataset, which represents the elementary knowledge, is first passed to a multi-expert decoder (MED). Each *expert* (*i.e.*, a pair of context module and classifier head) in the MED works on the dataset where pixels and their corresponding labels of dominant categories (*i.e.*, body and tail categories) in each image are masked. Together with other constraints, this expert-specific pixel-masking strategy enables the experts to reduce the impact of head categories and irrelevant pixels and acquire the contextual information of body or tail categories. The following Stage 2 deploys a multi-expert output ensemble (MOE) to ensemble the outputs of all experts from Stage 1 using decision weights. Instead of being user-specified, these weights are learned by a decision-maker to avoid the negative impact of the constraints. The proposed framework is model-agnostic and can be integrated with any popular semantic segmentation method, such as DeepLabv3+ (Chen et al., 2018), PSPNet (Zhao et al., 2017), OCRNet (Yuan et al., 2019) and so on.

To evaluate the MEDOE's performance in long-tailed semantic segmentation, especially for tail categories, we conduct theoretical and experimental analyses using the metrics of mean Intersection-over-Union (mIoU) and mean pixel accuracy (mAcc). They demonstrate the effectiveness of the proposed framework in boosting mAcc while keeping mIoU from decrease. In addition, we examine the potentials of MED and MOE despite that the information needed by the expert-specific pixel-masking strategy is unavailable in the inference phase of long-tailed semantic segmentation.

The key contributions of this paper can be summarized as follows:

- We conduct an empirical study to discover the problem of long-tailed distribution in semantic segmentation and reveal its significance.
- We advocate mAcc as a more important metric to evaluate the performance for body and tail categories in long-tailed semantic segmentation.
- We propose a model-agnostic multi-expert decoder and output ensemble framework that outperforms the several popular methods up to 2% in mIoU and 6% in mAcc on Cityscapes (Cordts et al., 2016) and ADE20K (Zhou et al., 2017) datasets.
- We demonstrate the ideal information referred to *Oracle* improves substantially the results, with an average gain of 5% in mIoU and 6% in mAcc on Cityscapes and 12% in mIoU and 19% in mAcc on ADE20K datasets. The results open a very promising direction for future research.

## 2 RELATED WORK

**Semantic Segmentation.** FCN (Long et al., 2015) is regarded as the pioneer in the field because of introducing the full convolution on the whole image and formulating the semantic segmentation task as per-pixel classification with the basic framework of *backbone-context module*. Subsequently, various advanced methods have been introduced based on FCN, they can be roughly divided into two directions. One is to design novel backbone for more robust feature representation (Wang et al., 2020a; Yu & Koltun, 2017). HRNet introduced a parallel backbone network to generate high-resolution representations. The other is to enrich contextual information for each pixel (Fu et al., 2019; He et al., 2019; Huang et al., 2019; Yuan et al., 2019). For instance, combining high-level feature and low-level feature to extract global information (Amirul Islam et al., 2017; Badrinarayanan et al., 2017), introducing large receptive field, such as dilated or atrous convolutions (Chen et al., 2018; Zhao et al., 2017) to gather multi-scale contextual cues and building the feature pyramids (Kirillov et al., 2019). However, these approaches ignore the impact of data distribution, and our work is the first to explicitly focus on the long-tailed data distribution in semantic segmentation.

**Long-tailed visual recognition. Re-balancing/Re-weighting:** The most widely-used and straightforward solution for long-tailed distribution is re-balancing the contribution of each category in the training phase. Re-weighting approaches (Cao et al., 2019b; Liu et al., 2019; Lin et al., 2017) adjust the loss function or boost larger weights on tail categories. Re-balancing approaches (He & Garcia, 2009; Kim et al., 2020; Chu et al., 2020) achieve data balance based on over-sampling the low-frequency categories, under-sampling the high-frequency categories or even data augmentation, by generating additional samples to complement tail categories. However, whether sample-wise or loss-wise re-balancing approaches increase the performance of minority categories while the accuracy of majority ones is compromised. **Ensembling and Grouping:** Recent studies show a new trend of overcoming the long-tailed problem by multi-expert or multi-branch strategy. BBN (Zhou et al., 2020) adopted two-branches to focus on normal and reversed sampling. LFME (Xiang et al., 2020), RIDE (Wang et al., 2020b) and ACE (Cai et al., 2021) trained on relatively balanced sub-groups, which separated training data by category frequency, and ensembled together into a multi-expert architecture. Those methods learn diverse classifiers parallel with knowledge distillation, distribution-aware expert selection or complementary experts. Despite being suffered from re-balance tricks, the success gained on image recognition has shown the great potential of these methods.

## 3 METHODS

Following the case described in Table 1, each category in semantic segmentation is tagged *head*, *body*, or *tail*, according to its rank based on the corresponding pixel frequency. In particular, body

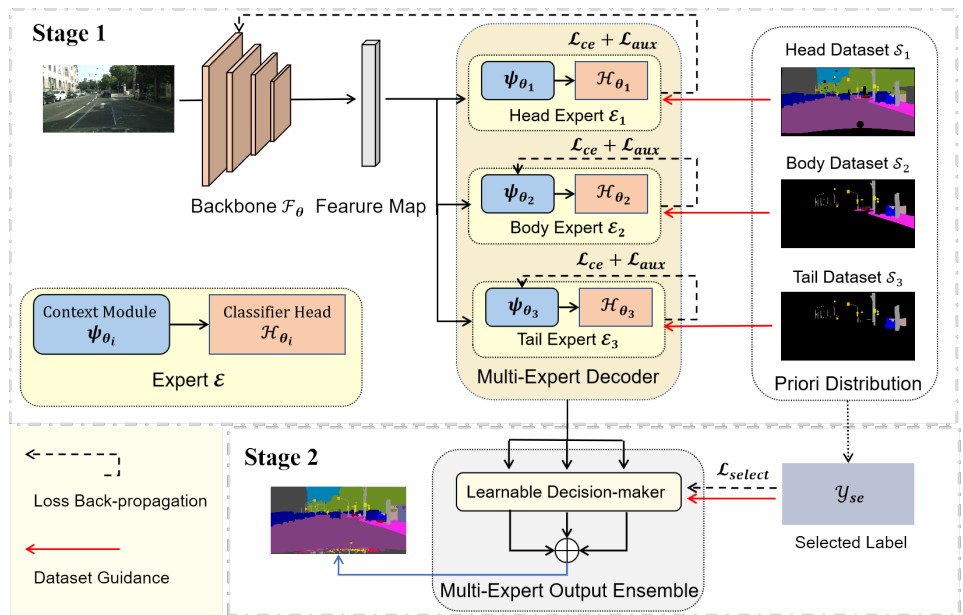

Figure 2: The overview of the proposed framework. **Stage 1**, Followed by shared backbone for representation training, the priori distribution allocate a specific masked dataset for each expert with the priori knowledge, then a MED module combined with context module and classifier self-adaptive generate contextual information on own dominating categories and classify pixels with $L_{ce}$ and $L_{aux}$. **Stage 2**, we generate the select label $y_{se}$ by priori distribution and MOE ensemble the outputs of all experts by a learnable decision-maker with the guidance of $y_{se}$.

categories refer to those ranked lower than the head but higher than the tail. Figure 2 provides an overview about the two-stage MEDOE framework. At Stage 1, given a low-level feature map extracted by a backbone from the input image, a multi-expert decoder (MED) is employed to discover contextual information, which has considerable importance in semantic segmentation. The MED includes several experts, each of which is composed of a context module and a classifier head, corresponding to head, body, and tail categories (respectively referred to as *head expert*, *body expert*, and *tail expert*). Following an expert-specific pixel-masking strategy, each expert processes a unique dataset with the image pixels of certain categories masked, which ensures the expert to focus on its corresponding categories and predict with higher confidence. Minimizing both Cross-Entropy (CE) and auxiliary loss functions, the head expert updates the backbone's parameters on the entire dataset, whereas the others refine their own context modules, in order to constrain the effect of interfering pixels and the distribution of masked data. At Stage 2, a multi-expert output ensemble (MOE) is designed to aggregate the outputs from different experts for predicting the label of each pixel labels. The weights used by MOE are from a learnable decision-maker, instead of those user-specified, so as to avoid the negative impact of the afore-mentioned constraints.

## 3.1 MED: MULTI-EXPERT DECODER

**The MED** is designed to draw out different contextual information from the feature map extracted by the backbone, in order to adjust the boundaries of classifiers, based on the observation that a model's performance in semantic segmentation tend to be superior for head categories. Specifically, the parameters of an expert are determined by its corresponding pixel-masking dataset, while the local optima of head categories can be avoided. The conventional context modules with large-scale receptive field convolutions, such as ASPP (Chen et al., 2018), PPM (Zhao et al., 2017) and OCR (Yuan et al., 2019), can self-adaptively generate different contextual information from dominant categories in the expert-specific pixel-masking datasets. This process can be defined as follows,

Given a training set $\mathbb{D} = \{(X, Y); C\}$, $X$ denotes the data and $Y$ denotes ground truth labels, with $C$ categories in total. Then we realign $C$ according to the categories frequency in pixel level, $C = \{1, 2, \ldots, c\}$, where $(F(i) > F(j), \text{ if } (i > j))$, $F(i)$ denotes the frequency of categories

*i.* For $K$ experts $\mathbb{E} = \{\mathcal{E}_1, \mathcal{E}_2, \ldots, \mathcal{E}_K\}$, we assign $K$ expert-specific masked datasets for different experts $\mathbb{S} = \{\mathcal{S}_1, \mathcal{S}_2, \ldots, \mathcal{S}_K\}$. In this paper, we set $K = 3$ by default, based on the intuition of head, body and tail categories, $c_b$ and $c_t$ denotes the first body and tail categories.

$$\mathcal{S}_1 = \{1, 2, \ldots, c\}, \quad \mathcal{S}_2 = \{c_b, c_{b+1}, \ldots, c\}, \quad \mathcal{S}_3 = \{c_t, c_{t+1}, \ldots, c\}. \tag{1}$$

In this case, the body and tail categories can be exposed to their dominant experts, and we also adopted some overlapped categories to expose fewer categories to more experts. For hard pixel segmentation, the overlapping strategy allows the experts who dominate the category to integrate the outputs of others. Followed the shared backbone $f_\theta$, experts have chance to extract contextual information focused on "experts" own categories by MED. To reduce overfitting in minority categories, we only adopt a small number of convolutions in context module $\Psi = \{\psi_{\theta_1}, \psi_{\theta_2}, \ldots, \psi_{\theta_K}\}$, and classifier head $\mathbb{H} = \{h_{\theta_1}, h_{\theta_2}, \ldots, h_{\theta_K}\}$. Finally, the expert $\mathcal{E}_i$ outputs prediction:

$$z_i = h_{\theta_i}(\psi_{\theta_i}(f_\theta(x))) \tag{2}$$

**Diverse data distribution-aware loss function.** We require each expert to perform well in their expert-specific masked dataset. Therefore, instead of constraining the final prediction as prior approaches, we apply the diverse data distribution-aware loss function on each separately. For each expert $\mathcal{E}_i$, we use the CE loss function to guide the pixel classification on the dataset $\mathcal{S}_i$.

$$L_{ce}(z_i, y; \theta_i) = -\sum_i^{\mathcal{S}_i} y_i \log\left(\text{softmax}(z_i)\right). \tag{3}$$

However, semantic segmentation trains each image end to end, and some categories were defined as interfering categories (IC). IC set $\mathcal{S}_i^{IC}$ refers to the pixels' categories not in $\mathcal{S}_i$, which means $\mathcal{S}_i^{IC} \cup \mathbb{S}_i = C$. Since we expect each expert to focus on the dominating categories, inevitably, $\mathcal{S}_i^{IC}$ is the main source of confusion to $\mathcal{E}_i$. We devise a novel suppression compensation measure by defining an auxiliary loss function to suppress the IC with an L2 regularization term. In addition to avoiding the over-confidence of confusing categories, for each expert $\mathcal{E}_i$, we minimize KL-divergence between the expert classification probability $\mathbf{p}$ with the reality label distribution $\mathbf{q}$ in ground truth $y$ over $\mathcal{S}_i$ categories.

$$L_{aux}(z_i, y; \theta_i) = \sum_{c_j}^{\mathcal{S}_i^{IC}} \left\| z_i^{c_j} \right\| + \sum_{c_j}^{\mathcal{S}_i} \mathbf{p}_{c_j} \log\left(\frac{\mathbf{p}_{c_j}}{\mathbf{q}_{c_j}}\right) \quad, \quad \mathbf{p_i} = \text{softmax}(z_i). \tag{4}$$

Overall, the diverse data distribution-aware loss function for expert $\mathcal{E}_i$ is

$$L(z_i, y; \theta_i) = L_{ce}(z_i, y; \theta_i) + \alpha L_{aux}(z_i, y; \theta_i), \tag{5}$$

where $\alpha$ is the hyper-parameter to balance the loss of $L_{ce}$ and $L_{aux}$. We empirically set $\alpha = 0.2$ by default. Since these dominant categories are complementary to each other, the $K$ experts learned at Stage 1 are good and distinctive from each other.

**Oracle case with known decision information.** As mentioned, the MED, loss terms and expert-specific pixel-masking strategy enable each expert to focus better on their dominant categories. For the intuitive assumption of the Stage 1, our model will achieve the theoretically optimal result when each pixel gets the prediction through the expert who dominates its category. We call this situation the "Oracle" case, and take such a case as the theoretical upper bound of our method. Specifically, due to the overlap strategy, when the ground truth label $y$ of pixel $x$ satisfied: $y \in \mathcal{S}_n \& y \notin \mathcal{S}_{n+1}$, we choose expert $\mathcal{E}_n$ to output the final classification probability $\mathbf{p}_O$ of instance $x$:

$$\mathbf{p}_O = \text{softmax}(h_{\theta_n}(\psi_{\theta_n}(f_\theta(x)))) \tag{6}$$

### 3.2 MOE: MULTI-EXPERT OUTPUT ENSEMBLE

**The MOE with a learnable decision-maker** is devised upon unavailable information on decisions, and aims to ensemble the probabilities for classification. The MOE takes as input the outputs from all experts at Stage 1, and makes a decision on label selection. For the n-th expert, if the ground truth label $y$ of a pixel $x$ belongs to the expert's dominant categories (*i.e.*, $y \in \mathcal{S}_n$), ideally the expert should be selected, $y_{se} = n$. We construct a feature embedding layer and a classifier to learn this decision-maker. Specifically, we concatenate the classification probabilities $\mathbf{p}$ for $x$ through the $K$ experts from first to k-th expert and project them to a scale by a 2D convolutional layer $\mathcal{W}_1$,

followed by ReLU, and finally, apply a k-th classifier $\mathcal{W}_2^k$ with a Softmax function to get selection probabilities **s** between each expert in [0,1]:

$$\mathbf{s}(\mathbf{x}) = \text{softmax}(\mathcal{W}_2^k(\text{ReLU}(\mathcal{W}_1[\mathbf{p}_1 \oplus \mathbf{p}_2 \oplus \cdots \oplus \mathbf{p}_k]))). \tag{7}$$

In the training phase, the decision-maker is optimized with a CE loss:

$$L_{select}\left(s(x), y_{se}\right) = -\sum_{i=1}^k y_{se}^i \log\left(s(x)_i\right). \tag{8}$$

In the inference phase, instead of a simply hard decision strategy, for instance $x$ we adopt the select probability $s$ as a soft decision weight $\mathbf{W}$ for all experts, the final classification probabilities:

$$\mathbf{p}_{final} = \frac{1}{k}\sum_{i=1}^k \mathbf{w}_i \mathbf{p}_i \quad , \quad [\mathbf{w}_1 \oplus \mathbf{w}_2 \oplus \cdots \oplus \mathbf{w}_k] = \mathbf{W} \tag{9}$$

### 3.3 METRIC THEORETICAL ANALYSIS

**Theoretical analysis between mIoU and mAcc in Semantic Segmentation.** Due to the experimental results on Cityscapes, in Sec 4, we were surprised to find that the mAcc made a great improvement while the mIoU remained stable. This misalignment inspires us to explore the correlation between mIoU and mAcc in long-tailed semantic segmentation because we use mIoU as the primary evaluation metric in segmentation, but traditional long-tailed recognition adopts mAcc as a standard metric. We refer $TP$, $FN$ and $FP$ to true positives, false negatives and false positives. $Acc_i$ and $IoU_i$ refer to pixel accuracy and Intersection-over-Union of category $i$ can be present as,

$$Acc_i = TP_i/(TP_i + FN_i), \quad IoU_i = TP_i/(TP_i + FN_i + FP_i). \tag{10}$$

The correlation between $FP$ and $FN$ refers to : $\sum_{i=1}^c FP_i = \sum_{i=1}^c FN_i$. Meanwhile, $\sum_{i=1}^c FN_i$ is more affected by the incorrect instances in dominant head categories than the tail categories (See **Appendix** for details). Therefore, when evaluating the IoU on tail categories, the $FN$-related $FP$ will be suppressed by head categories, and make IoU remain stable. We believe that mIoU is a metric to focus on the segmentation results of whole images, which will be affected by head categories. Different from mIoU, the terms in mAcc formula are correlated to their own categories, which is a more fair metric of the improvement of body and tail categories. Thus, we regarded mAcc as a relatively important metric in our experiments. Meanwhile, consider a situation where, in pursuit of accuracy growth, we classify all the pixels nearby the tail category instances into this category, resulting in a substantial increase about $FP_i$, which is harmful for segmentation. According to **Remark 1**, only satisfies the precondition of no significant drop in mIoU, the mAcc increase does not cause the harm mentioned previously and tends to better performance in tail categories.

## 4 EXPERIMENTS

We conducted the experiments on two challenging benchmark datasets, by integrating our MEDOE framework into DeepLabv3+, PSPNet, OCRNet and Segformer to evaluate the effectiveness. Due to the limited space, benchmarks & implementation details are left in the **Appendix**.

### 4.1 COMPARISON RESULTS

**Cityscapes.** To validate the flexibility and efficiency of our MEDOE framework, we experiment with our method on Cityscapes. As shown in Table 2 and 4, compared with the current methods, MEDOE significantly improved mAcc without reducing mIoU and too many extra parameters. Specifically, for ResNet-50c, ResNet-101c, HRNet-W48 and MIT-B3 (Transformer-based), there are 3.36%, 3.46%, 1.03% and 2.88% gains in mAcc for DeepLabv3+, PSPNet, OCRNet and Seg-Former. Especially, our method achieved more impressive results than the ordinary re-weighting method (Focal Loss). This is because Focal Loss through the simple re-weighting method slightly improves the accuracy of the tail categories with a significant decrease in the performance of the head categories, which will cause an unacceptable decline in mIoU.

**ADE20K.** ADE20K is a challenging benchmark due to its various image scales and plenty of semantic categories in reality, which cause the long-tailed distribution. In this scenario, our method shows better performance than the current methods. The experimental results are summarized in

Table 2: Comparison of performance and memory size (Params) on the validation set of Cityscapes with current methods. †: Oracle results for the ideal scenario.

| Methods | Backbone | mIoU (%) | mAcc (%) | Params (GBs) |
|---|---|---|---|---|
| DeepLabv3+ (Chen et al., 2018) | ResNet-50c | 80.37 | 86.68 | 24.16(1.0×) |
| DeepLabv3+-MEDOE | ResNet-50c | 80.20(−0.17) | 90.04(+3.36) | 34.31(1.4×) |
| DeepLabv3+-MEDOE† | ResNet-50c | 84.14(+3.77) | 92.38(+5.70) | 34.31(1.4×) |
| PSPNet (Zhao et al., 2017) | ResNet-50c | 78.32 | 85.52 | 18.33(1.0×) |
| PSPNet-MEDOE | ResNet-50c | 78.02(−0.30) | 88.98(+3.46) | 20.37(1.1×) |
| PSPNet-MEDOE† | ResNet-50c | 84.32(+4.56) | 91.90(+5.34) | 20.37(1.1×) |
| DeepLabv3+ | ResNet-101c | 80.67 | 87.58 | 25.02(1.0×) |
| DeepLabv3+-MEDOE | ResNet-101c | 80.23(−0.44) | 91.12(+3.52) | 35.03(1.4×) |
| DeepLabv3+-MEDOE† | ResNet-101c | 84.51(+3.84) | 92.63(+5.05) | 35.03(1.4×) |
| PSPNet | ResNet-101c | 79.76 | 86.56 | 20.11(1.0×) |
| PSPNet-MEDOE | ResNet-101c | 79.79(+0.03) | 90.88(+4.32) | 23.81(1.1×) |
| PSPNet-MEDOE† | ResNet-101c | 84.32(+4.56) | 91.90(+5.34) | 23.81(1.1×) |
| OCRNet (Yuan et al., 2019) | HRNet-W48 | 80.70 | 88.11 | 23.44(1.0×) |
| OCRNet-MEDOE | HRNet-W48 | 80.21(−0.49) | 89.14(+1.03) | 30.27(1.3×) |
| OCRNet-MEDOE† | HRNet-W48 | 85.29(+4.59) | 93.38(+5.27) | 30.27(1.3×) |
| SegFormer (Xie et al., 2021) | MIT-B3 | 81.94 | 88.28 | 30.02(1.0×) |
| SegFormer-MEDOE | MIT-B3 | 81.67(−0.27) | 91.16(+2.88) | 38.65(1.3×) |
| SegFormer-MEDOE† | MIT-B3 | 85.67(+3.73) | 93.49(+5.21) | 38.65(1.3×) |

Table 3: Comparison of performance and memory size (Params) on the validation set of ADE20K with with current methods. †: Oracle results for the ideal scenario.

| Methods | Backbone | mIoU (%) | mAcc (%) | Params (GBs) |
|---|---|---|---|---|
| DeepLabv3+ (Chen et al., 2018) | ResNet-50c | 42.11 | 54.13 | 25.82(1.0×) |
| DeepLabv3+-MEDOE | ResNet-50c | 43.82(+1.71) | 60.02(+5.89) | 35.49(1.4×) |
| DeepLabv3+-MEDOE† | ResNet-50c | 53.34(+11.23) | 73.97(+19.84) | 35.49(1.4×) |
| PSPNet (Zhao et al., 2017) | ResNet-50c | 40.46 | 51.42 | 19.81(1.0×) |
| PSPNet-MEDOE | ResNet-50c | 41.76(+1.30) | 54.27(+2.85) | 23.49(1.1×) |
| PSPNet-MEDOE† | ResNet-50c | 52.00(+11.54) | 71.61(+20.19) | 23.49(1.1×) |
| DeepLabv3+ | ResNet-101c | 44.60 | 56.28 | 26.31(1.0×) |
| DeepLabv3+-MEDOE | ResNet-101c | 46.13(+1.42) | 61.12(+4.84) | 36.27(1.4×) |
| DeepLabv3+-MEDOE† | ResNet-101c | 55.18(+10.58) | 76.82(+20.54) | 36.27(1.4×) |
| PSPNet | ResNet-101c | 43.33 | 54.51 | 22.94(1.0×) |
| PSPNet-MEDOE | ResNet-101c | 44.31(+0.98) | 59.85(+5.19) | 25.69(1.1×) |
| PSPNet-MEDOE† | ResNet-101c | 53.47(+10.14) | 72.86(+18.45) | 25.69(1.1×) |
| OCRNet (Yuan et al., 2019) | HRNet-W48 | 42.53 | 54.91 | 33.03(1.0×) |
| OCRNet-MEDOE | HRNet-W48 | 43.31(+0.78) | 58.96(+4.05) | 37.67(1.1×) |
| OCRNet-MEDOE† | HRNet-W48 | 51.99(+9.46) | 74.68(+19.77) | 37.67(1.1×) |
| SegFormer (Xie et al., 2021) | MIT-B3 | 47.13 | 60.84 | 30.02(1.0×) |
| SegFormer-MEDOE | MIT-B3 | 48.22(+1.09) | 64.06(+3.22) | 33.96(1.3×) |
| SegFormer-MEDOE† | MIT-B3 | 56.49(+9.36) | 79.41(+18.57) | 33.96(1.3×) |

Table 3 and 4, our method achieved 1.71%, 1.30%, 0.78% and 1.09% gains in mIoU and 5.89%, 2.85%, 4.05% and 3.22% gains in mAcc for DeepLabv3+, PSPNet, OCRNet and SegFormer. Furthermore, we illustrated the qualitative results in Figure 4 on the validation set of ADE20K. We can see our method achieved better segmentation results than baseline on these tail categories in different scenarios, such as pillow and lamp, which shows the consistent effectiveness of our method.

**Results in diverse category data subsets.** As illustrated in Table 4, our method achieved impressive mAcc gains for body and tail data subsets on Cityscapes and ADE20K. Specifically, we achieved overall improvement on ADE20K. For further, Category-wise performance improved comparisons between MEDOE and baseline is shown in Figure 3. MEDOE achieved both IoU and Acc improvements on Cityscapes and ADE20K benchmark datasets in almost all categories. Especially, our method has significant advantages for body and tail categories data subsets, which verifies our motivation that MED can extract effective contextual information to help pixel classify.

**Oracle results.** We investigated the ideal scenario to provide the exact experts' selecting decisions in the inference phase substitute the MOE, such that $\mathbf{W} = \mathbf{W}^*$. Reported results in this ideal scenario, referred to as *Oracle* in Table 2 and 5, shown impressive improvements in mIoU and mAcc over both current methods, with a consistent gain across all datasets. Specifically, our method gains 3%-4%

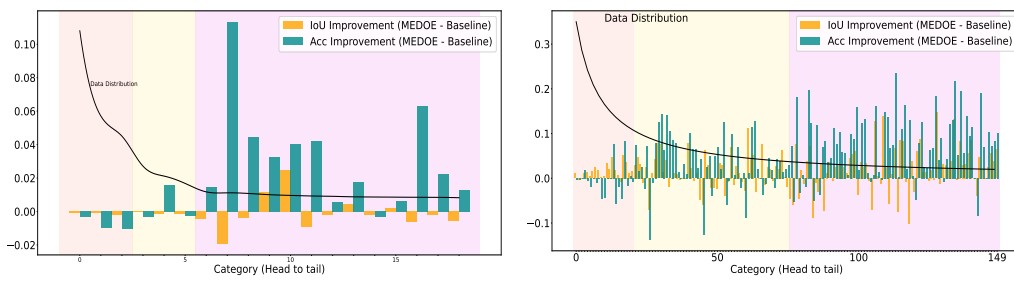

(a) Performance improvement on Cityscapes    (b) Performance improvement on ADE20K

Figure 3: Comparisons of each category performance on Cityscapes and ADE20K with MEDOE and baseline. Our MEDOE gains both IoU and Acc improvements in minority categories on Cityscapes and ADE20K, as well as avoid decreasing the majority performance.

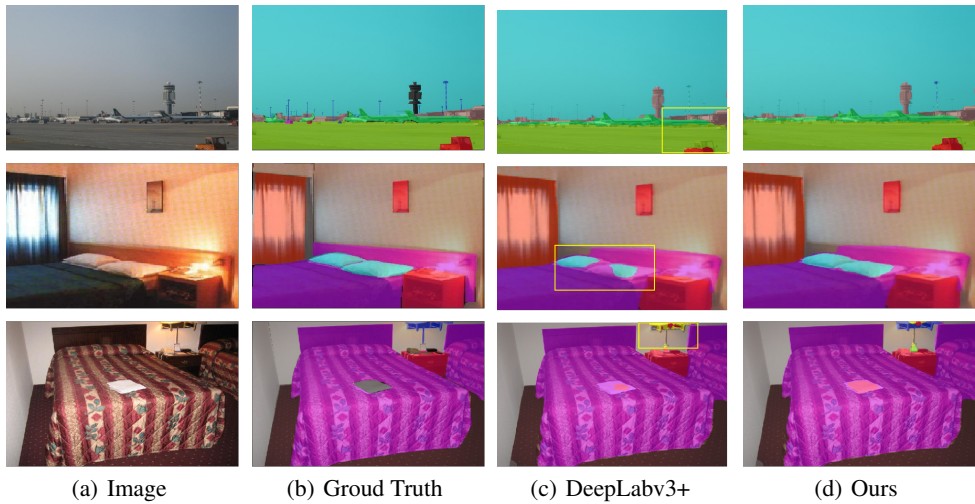

(a) Image          (b) Groud Truth          (c) DeepLabv3+          (d) Ours

Figure 4: Qualitative Visualization on the validation set of ADE20K with ResNet-50c as backbone. All the models here are trained under the same setting.

in mIoU and 5%-7% in mAcc on Cityscapes and particularly the improvements raise to 11%-13% in mIoU and 19%-22% in mAcc on ADE20K. We believe these impressive improvements convey several important messages. 1) It proves that our components in Stage 1 have trained a series of robust and effective experts which focused on their dominant categories. 2) Compared to the results of our method, they indicate that there exists a strategy building on top of MED that can largely outperform state-of-the-art methods. This suggests that MED has significant prospects, more efforts could be directed towards to constrain the expert decision and opening a door to promising avenues.

### 4.2 ABLATION STUDY

**Ablation of MED.** To illustrate that our MED self-adaptively extracts different contextual information guided by diverse masked datasets, indeed, rather than attributing entirely to the modifying classifiers' boundaries operation or simply increasing extra parameters (*i.e.* model-ensemble method). We designed **M**ulti-**C**lassifier network (**MC**) with a shared backbone and context module and model-ensemble method for a comparison experiment in the whole datasets on Cityscapes, the results are shown in Table 5. Just modifying the classifier boundaries is not effective in the tail categories, and even reduces the overall performance. Since the task is really difficult, the model-ensemble method simply enhancing extra parameters (multi-backbones) can not make overall improvements. Without our training strategies, it improves head performance but ignores the tail categories. Thus MED helps the network better classify pixels by different contextual information.

**Ablation of MOE.** As we mentioned, we introduced various user-specified output ensemble methods. For each pixel instance $x$, (2) Softmax method get the output $z_i$ of expert $\mathcal{E}_i$ from 1 to k-th and pixel category $c_i == j$ if $(\text{softmax}(z_i)) > \beta$, $\beta$ is set as 0.3, (3) Argmax method determine

Table 4: Comparison of performance (%) on each data subsets from Cityscapes and ADE20K with DeepLabv3+ and ResNet-50c. †: Baseline with Focal Loss, ‡: Oracle results for the ideal scenario.

| Methods | All | | Head | | Body | | Tail | |
|---|---|---|---|---|---|---|---|---|
| | mIoU | mAcc | mIoU | mAcc | mIoU | mAcc | mIoU | mAcc |
| Cityscapes | **80.37** | 86.68 | **95.26** | **97.74** | **83.42** | 91.49 | 73.01 | 81.43 |
| Cityscapes† | 76.23 | 85.00 | 91.18 | 90.97 | 80.55 | 90.33 | 68.79 | 84.56 |
| Cityscapes-MEDOE | 80.20 | **90.04** | 95.03 | 96.52 | 83.39 | **94.25** | **73.65** | **86.15** |
| Cityscapes-MEDOE‡ | *84.14* | *92.38* | *96.43* | *97.72* | *87.12* | *96.70* | *78.59* | *88.88* |
| ADE20K | 42.11 | 54.13 | 65.50 | **78.71** | 44.75 | 59.75 | 33.64 | 42.97 |
| ADE20K† | 37.61 | 55.29 | 60.01 | 74.89 | 41.96 | 61.28 | 30.28 | 46.95 |
| ADE20K-MEDOE | **43.82** | **60.02** | **66.99** | 77.90 | **46.32** | **63.52** | **35.38** | **52.33** |
| ADE20K-MEDOE‡ | *53.34* | *73.97* | *67.38* | *75.49* | *55.65* | *74.84* | *46.75* | *74.09* |

the outputs according to the confidence of the maximum softmax on each expert, (4) Group average method ensemble $z_i$ of $\mathcal{E}_i$ by grouping average weight. Comparisons between MOE with the above-mentioned methods in Table 6. Our MOE achieved improvements overall.

Table 5: Ablation of **MC** network , model-ensemble method and **MED** (ours) with DeepLabv3+ and ResNet-50c on Cityscapes to verify the effectiveness of *multi-context decoder* module. Results are reported in mIoU (%), mAcc (%) and Params (GBs)

| Methods | All | | Many | | Medium | | Few | | Params |
|---|---|---|---|---|---|---|---|---|---|
| | mIoU | mAcc | mIoU | mAcc | mIoU | mAcc | mIoU | mAcc | |
| Baseline | 42.11 | 54.13 | 65.50 | 78.71 | 44.75 | 59.75 | 33.64 | 42.97 | 25.82(1.0×) |
| +MC | 40.15 | 53.79 | 62.01 | 75.51 | 42.53 | 56.20 | 32.15 | 45.18 | 28.39(1.1×) |
| +Ensemble | 42.50 | 54.46 | 65.59 | 78.62 | 45.00 | 60.31 | 33.87 | 42.95 | 77.98(3.0×) |
| +MEDOE | 43.82 | 60.02 | 66.99 | 77.90 | 46.32 | 63.52 | 35.38 | 52.33 | 35.49(1.4×) |

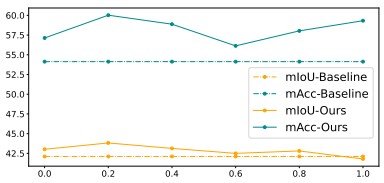

| Aggregation | mIoU | mAcc |
|---|---|---|
| MOE | 80.20 | 90.04 |
| Softmax(2) | 72.31 | 82.32 |
| Argmax(3) | 70.45 | 83.14 |
| Group Avg(4) | 78.87 | 86.49 |

Figure 5: Ablation w/ or w/o Auxiliary Loss and the hyper-parameter $\alpha$ on ADE20K with DeepLabv3+ and ResNet-50c

Table 6: Ablation study on various outputs ensemble methods with DeepLabv3+ and ResNet-50c on Cityscapes.

**Effectiveness of the auxiliary loss function.** An ablation with or without an auxiliary loss function and the hyper-parameter $\alpha$ on ADE20K was conducted to prove the motivation that we introduce auxiliary loss function. The results in Figure 5 shown that in the case of using CE loss function alone (Ours with $\alpha$=0.0), our method can achieve the improvement compared to baseline, but the improvement is limited. Meanwhile, when the hyper-parameter $\alpha$=0.2 we gain the best performance. Training without the auxiliary loss function constraint is more likely to be confused by the unavoidable IP and decrease the performance overall. The introduction of auxiliary loss function obtains a more consistent and stable improvement.

## 5 CONCLUSION

In this paper, we investigate and identify the long-tailed distribution in semantic segmentation, and motivated by the fact that existing methods ignore this issue and the performance declines for tail categories. We proposed a MEDOE framework to overcome this challenging problem. The MED, MOE module and a series of strategies (*e.g.*, auxiliary loss function and priori distribution) are introduced to help each expert extract different contextual information and promote the performance in dominant categories. Our method achieved great improvements across different current methods up to 2% gains in mIoU and 6% gains in mAcc on Cityscapes and ADE20K. Furthermore, training a robust output ensemble decision-maker appears to be a very promising way of constraining the inference, as demonstrated with the significantly improved results obtained by the oracle.

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

# A APPENDIX

## A.1 BENCHMARKS & IMPLEMENTATION DETAILS

**Benchmarks: Cityscapes** (Cordts et al., 2016) is a dataset that focuses on semantic understanding of urban street scenes and contains 19 semantic classes. It contains 5K annotated images with pixel-level fine annotations and 20K coarsely annotated images. The finely annotated 5K images are split into sets with numbers 2975, 500 and 1525 for training, validation and testing.

**ADE20K** (Zhou et al., 2017) is a challenging scene parsing dataset. It contains 150 categories and diverse scenes with 1038 image-level labels. The training and validation sets consist of 20K and 2K images, respectively.

**Implementation Details.** We initialize backbones with the weights pre-trained on ImageNet. For DeepLabv3+, we using CNN-based ResNet-50c and ResNet-101c as backbones, which switch the first 7×7 convolution layer to three 3×3 convolutions. HRNet-W48 (Wang et al., 2020a) is adopted for OCRNet. MIT-B3 as a Transformer-based backbone is adopted for SegFormer (Xie et al., 2021), both of which are popular in semantic segmentation. For CNN-based network, we use SDG and poly learning rate schedule (Zhao et al., 2017) with factor $\left(1 - \frac{iter}{total_{iter}}\right)^{0.9}$. The initial learning rate is set as 0.01 and weight decay is 0.0005. We adopt AdamW with $6 \times 10^{-5}$ learning rate and 0.01 weight decay. We set the image crop size to 512×1024, batch size as 8 and training iterations as 80K on Cityscapes by default. For ADE20K, the crop size of images is set as 512×512, the batch size is set as 16 and training iterations are set as 80K if not stated otherwise. In the training phase, we augment data samples with the standard random scale in the range of [0.5, 2.0], random horizontal flipping, random cropping, as well as random color jittering. For inference, the input image size of ADE20K is the same as the size during training, but for Cityscapes the input image is scaled to 1024×2048, no tricks(*e.g.*multi-scale with flipping) will be adopted during testing. All experiments are implemented on the Nvidia A6000.

## A.2 ADDITIONAL PROOF OF THEORETICAL ANALYSIS BETWEEN MIOU AND MACC

Before we proceed with the additional proof, we introduce some formulas for mAcc and mIoU.

$$mAcc = \frac{1}{C}\sum_{i=1}^{C} Acc_i = \frac{1}{C}\sum_{i=1}^{C} \frac{TP_i}{TP_i + FN_i},$$

$$mIoU = \frac{1}{C}\sum_{i=1}^{C} IoU_i = \frac{1}{C}\sum_{i=1}^{C} \frac{TP_i}{TP_i + FN_i + FP_i}. \tag{11}$$

$$\sum_{i=1}^{C} FP_i = \sum_{i=1}^{C} FN_i \tag{12}$$

$$num_i = TP_i + FP_i \tag{13}$$

### A.2.1 ADDITIONAL PROOF OF MACC IS THE BETTER TAIL-SENSITIVE METRIC

According to the prior information from experimental statistics, the proportions of pixel instances of the head, body, and tail categories in the semantic segmentation are: 80%, 15% and 5%. We analyzed the items in the mIoU and mAcc formulas and found that the difference between them was the $FP_i$. According to Eq. 12 We derive

$$\sum_{i=1}^{C} FP_i = \sum_{i=1}^{C} FN_i = \sum^{C_h} FN_i + \sum^{C_b} FN_i + \sum^{C_t} FN_i$$

$$\approx \sum^{C_h}(1 - Acc_i) \times num_i + \sum^{C_b}(1 - Acc_i) \times num_i + \sum^{C_t}(1 - Acc_i) \times num_i \tag{14}$$

$$= (1 - mAcc_h)\sum^{C_h} num_i + (1 - mAcc_b)\sum^{C_b} num_i + (1 - mAcc_t)\sum^{C_t} num_i$$

$$= (1 - mAcc_h) \times 0.8NUM + (1 - mAcc_b) \times 0.15NUM + (1 - mAcc_t) \times 0.05NUM$$

where, $h$, $b$ and $t$ refer to head, body, tail categoris and $NUM$ refers to the pixel instance number of overall datasets.

From Eq. 14, we obtain the conclusion that due to the large instance base of the head categories, $FN_i$ and $FN$-related $FP_i$ items are dominated by head categories. Therefore, the $IoU_i$ of each category $i$ and $mIoU$ will be dominated by head categories because of the item $FP_i$.

Instead of $mIoU$, for each category $i$ of $mAcc$, the items of $Acc_i$ are only related to its own category $i$, and $mAcc$ will not be dominated by the head categories. Thus $mAcc$ is a fair and tail-sensitive metric.

### A.2.2 ADDITIONAL PROOF OF REMARK 1

To better understand the correlation between mean IoU and mean Acc in long-tailed semantic segmentation and **Remark 1**. According to the equation Eq. 10 and the precondition from the experiments. The category Acc and IoU in our method become :

$$
\begin{aligned}
I\hat{o}U_i &\approx IoU_i, \\
\hat{Acc}_i &= (1+p)ACC_i.
\end{aligned}
\tag{15}
$$

And it is clear from Eq.13, We obtain the result:

$$
\hat{Acc}_i = (1+p)ACC_i
$$

$$
\Rightarrow \hat{Acc}_i = \frac{\hat{TP}_i}{\hat{TP}_i + \hat{FN}_i} = \frac{\hat{TP}_i}{TP_i + FN_i} = (1+p)\frac{TP_i}{TP_i + FN_i} = (1+p)ACC_i
\tag{16}
$$

$$
\Rightarrow \hat{TP}_i = (1+p)TP_i
$$

$$
I\hat{o}U_i \approx IoU_i \Rightarrow I\hat{o}U_i = \frac{\hat{TP}_i}{\hat{TP}_i + \hat{FN}_i + \hat{FP}_i} = \frac{\hat{TP}_i}{TP_i + FN_i + \hat{FP}_i} \approx \frac{TP_i}{TP_i + FN_i + FP_i} = IoU_i
$$

$$
\Rightarrow (1+p)TP_i \times (TP_i + FN_i + FP_i) = TP_i \times (TP_i + FN_i + \hat{FP}_i)
$$

$$
\Rightarrow (1+p)TP_i \times (num_i + FP_i) = TP_i \times (num_i + \hat{FP}_i)
$$

$$
\Rightarrow \Delta FP_i = p \times (num_i + FP_i)
\tag{17}
$$

$$
Acc_i = \frac{TP_i}{TP_i + FN_i} \ \& \ IoU_i = \frac{TP_i}{TP_i + FN_i + FP_i} \Rightarrow FP_i = \frac{Acc_i}{IoU_i} \times num_i - num_i
\tag{18}
$$

where $\Delta FP_i$ is the increased false positive from baseline to our method.

To guarantee the classifier effective and segment more tail categories, it should satisfied:

$$
\Delta FP_i = p \times (num_i + FP_i) \ll num_i
\tag{19}
$$

We observe that the results in Cityscapes benchmark has achieved a high value, $e.g. IoU_i = 0.8$ and $Acc_i = 0.85$. According to this precondition and Eq. 18, $FP_i = 0.0625 \times num_i$ and $p$ are both small value, which means $\Delta FP_i$ is a minimum value and satisfied Eq. 19. This is just the reason that Acc improved while IoU not decrease is significant for tail categories segmentation.

### A.3 ADDITIONAL PROOF OF THE CORRELATION BETWEEN PERFORMANCE AND DATA DISTRIBUTION

As shown of Figure 6, the baseline method on Cityscapes and ADE20K perform not well on certain categories, and we can clearly see that these categories mainly fall in the tail and body data subsets. This further demonstrates that the long-tailed data distribution limits the overall performance of the baseline method by constraining the accuracy of certain categories.

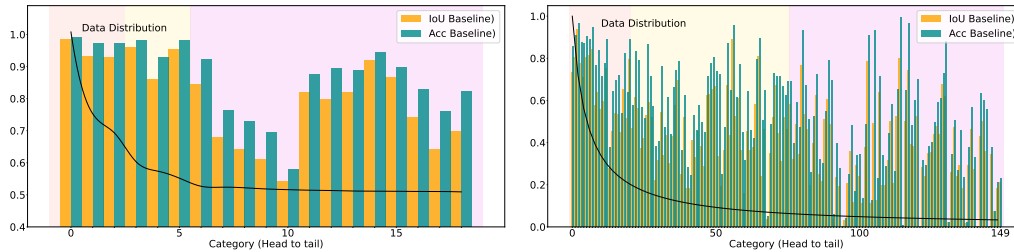

(a) Data distribution and performance with baseline on Cityscapes.

(b) Data distribution and performance with baseline on ADE20K.

Figure 6: The existence of long-tailed distribution in semantic segmentation, we set baseline as DeepLabv3+ and ResNet-50c by default. (a) and (b) shown the data distributions on Cityscapes and ADE20K are long-tailed, which cause the better performance on the majority categories yet suppress the minority categories. It should be noted that we reordered the categories in Cityscapes according to Pixel Frequency.

## A.4 ABLATION WITH LONG-TAILED METHODS

Table 7: Pearson coefficients between categories frequency and accuracy, the lower value means much weaker correlation.

| Dataset | $\rho_{X,Y}(\%)$ |
|---|---|
| CIFAR-100 | 75.9 |
| ADE20K-pixel | 36.8 |
| Cityscapes-pixel | 58.9 |

In this section, we first calculate the Pearson correlation coefficient to show the correlation between category accuracy and category frequency (image category frequency on CIFAR-100) in Table 7. The weak correlation between category accuracy and pixel level frequency on Cityscapes and ADE20K causes challenge to re-weighting in semantic segmentation, which can be demonstrated in Table 8. We compared our work with traditional long-tailed classification methods to further explore the contribution of our work. We adopt the re-weighting method to modify the loss function and put larger weights on tail categories. Despite improving mAcc, the re-weighting method caused a lot of decrease in mIoU. For re-sampling, the contextual information is corrupted resulting in segmentation metrics at a very low level. In general, the current Pixel level re-balance approaches can not work well with the long tail distribution on semantic segmentation.

Table 8: Ablation with our method and pixel level re-weighting and re-sampling, all networks adopt ResNet-50c backbone.

| Benchmarks | Methods | mIoU | mAcc |
|---|---|---|---|
| Cityscapes | Baseline | 80.37 | 86.68 |
| | Reweighting-FocalLoss (Lin et al., 2017) | 76.23(−4.14) | 85.00(−1.68) |
| | Reweighting-LDAMLoss (Cao et al., 2019a) | 77.52(−2.85) | 85.15(−1.43) |
| | Reweighting-SeesawLoss (Wang et al., 2021) | 67.58(−12.79) | 74.35(−12.33) |
| | Re-sampling | 66.79(−13.58) | 75.21(−11.47) |
| | Baseline+MED | 80.20(−0.17) | 90.04(+3.36) |
| ADE20K | Baseline | 42.11 | 54.13 |
| | Reweighting-FocalLoss (Lin et al., 2017) | 37.61(−4.50) | 55.29(+1.16) |
| | Reweighting-LDAMLoss (Cao et al., 2019a) | 40.39(−1.72) | 48.78(−5.35) |
| | Reweighting-SeasawLoss (Wang et al., 2021) | 33.87(−8.24) | 40.19(−13.94) |
| | Resampling | - | - |
| | Baseline+MED | 43.82(+1.71) | 60.02(+5.89) |

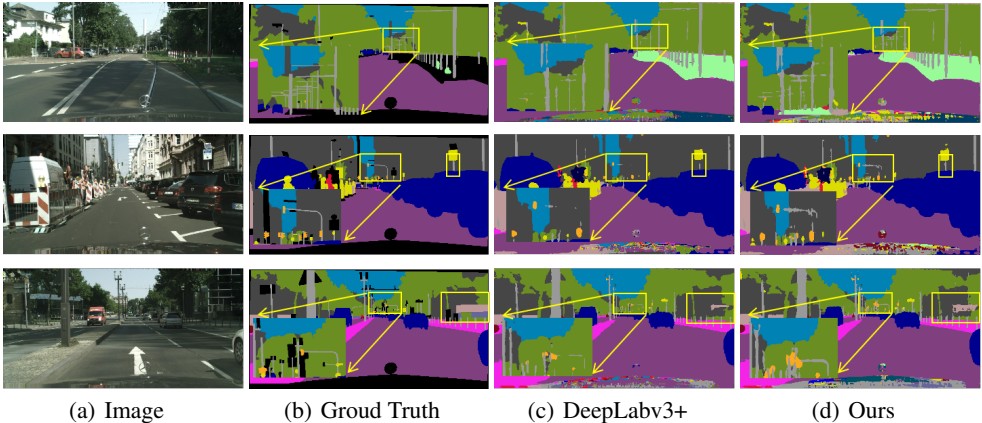

| (a) Image | (b) Groud Truth | (c) DeepLabv3+ | (d) Ours |

Figure 7: Qualitative Visualization results on the validation set of Cityscapes with ResNet-50c as backbone. All the models here are trained under the same setting.

## A.5 ANALYSIS OF THE PERFORMANCE GAP BETWEEN CITYSCAPES AND ADE20K

According to the experiments' results of Sec4, our method has achieved impressive performance in both mIoU and mAcc on ADE20K dataset. However, there seems to be a gap between the performance on Cityscapes and ADE20K. We analyze the causes: 1) Compared to ADE20K, there are fewer categories in Cityscapes and a more pronounced long-tail distribution (higher proportion of head category instances), so it is easier to fall into local optimum when training body and tail experts, resulting in the overfitting of these categories. Finally, it caused that the increase of $FN_{ht}$ and $FP_{ht}$ on the overall datasets. $FN_{ij}$ and $FP_{ij}$ denote to the pixel instance which the ground truth belongs to category $i$ but the prediction is $j$. 2) The performance of baseline methods on Cityscapes is at a high level. According to the above two reasons, our method improves the overall $TP$ on Cityscapes, but it will increase $FN_{ht}$ of the head categories and $FP_{ht}$ of the tail categories, respectively, resulting in $mIoU$ being at a relatively stable value.

From the macro level of image visualization, as shown in Figure 7, our method segments the surrounding part of the head categories into tail categories while segmenting the tail categories. We believe that such head accuracy decreases are acceptable and meaningful in real-world scenarios.

## A.6 QUANTITATIVE VISUALIZATION COMPARISONS ON CITYSCAPES AND ADE20K

In this section, we demonstrate the better performance of MEDOE framework with quantitative visualization on Cityscapes and ADE20K shown in Figure 7 and 8. We adopt ResNet-50c as backbone and all models trained under the same setting. In most semantic scenarios, our MEDOE method can achieve better performance in segmenting tail categories.

## A.7 ADDITIONAL EXPLANATIONS

### A.7.1 CONTEXTUAL MODULE

**Contextual module**, as an important module in semantic segmentation, refers to the extraction and aggregation of contextual information for pixels through a series of operations (*i.e.*, feature pyramids, atrous convolution, large-scale convolutional, attention mechanisms, and global pooling). Existing contextual modules include: ASPP (Chen et al., 2018), PSP (Zhao et al., 2017), or non-local (Wang et al., 2017).

**Contextual information** means the relationship between this pixel and the surrounding pixels and global information is regarded as contextual information. **The reasons why segmentation needs contextual module.** When solving semantic segmentation tasks if each pixel considers only its deep features, such as texture and color, it will be difficult to classify into the correct category (*i.e.* the

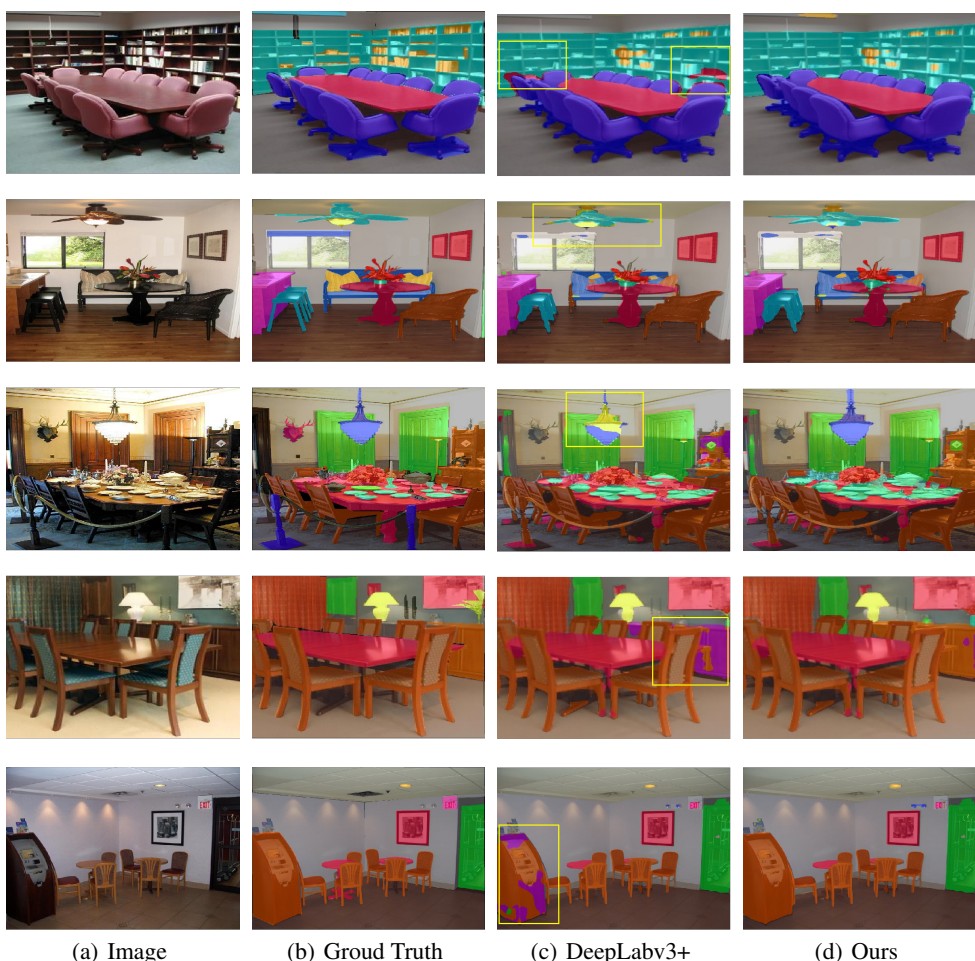

(a) Image      (b) Groud Truth      (c) DeepLabv3+      (d) Ours

Figure 8: Qualitative Visualization on the validation set of ADE20K with ResNet-50c as backbone. All the models here are trained under the same setting.

deep features of leaves and green grass will be very similar). At the same time, as stated in the work, semantic segmentation believes that each pixel is not I.I.D., but is related to the surrounding pixels. The correlation information can better help the segmentation task (*i.e.*the leaves are surrounded by branches, and the grass is likely to be surrounded by roads).

## A.8   ABLATION OF DIFFERENT EXPERTS

To measure the goodness and distinctiveness, We adopt the mAcc and per-category bias in Table 9 as the metrics to describe each expert. It seems to demonstrate that the *K* experts learned at Stage 1 are good and distinctive from each other, rather than simply boosting confidence through multi-expert training.

Furthermore, according to the well-known bias-variance decomposition (Wang et al., 2020b), per-category bias denotes:

$$\text{Error}(x; h) = E\left[(h(x; D) - Y)^2\right] = \text{Bias}(x; h) + \text{Variance}(x; h) + \text{ irreducible error }(x), \quad (20)$$

Table 9: Ablation of each expert with the mean accuracy and per-category bias on Cityscapes with Deeplabv3+ and ResNet-50c. The higher mean accuracy and lower bias are better.

|          | Many | | Medium | | Few | |
|----------|------|------|------|------|------|------|
|          | mAcc | bias | mAcc | bias | mAcc | bias |
| Overall  | 0.96 | 0.10 | 0.88 | 0.22 | 0.81 | 0.30 |
| Expert 1 | 0.97 | 0.09 | 0.86 | 0.24 | 0.75 | 0.36 |
| Expert 2 | -    | -    | 0.95 | 0.09 | 0.86 | 0.19 |
| Expert 3 | -    | -    | -    | -    | 0.91 | 0.13 |

### A.9   LONG-TAILED SEMANTIC SEGMENTATION

To the best of our knowledge, there is a contemporaneous (Cui et al., 2022) work with our paper. Although, we almost simultaneously focus on the long-tailed distribution as an important reason for constraining semantic segmentation performance, there are some differences between our work and Region rebalance (Cui et al., 2022):

**Perspective difference:** Region rebalance was concerned about solving the problem of category rebalance, while our work was more focused on improving the recognition of tail and body categories and placed special emphasis on the significance of segmenting the body, and tail categories.

**Methods difference:** Region rebalance relieved the categories imbalance with an auxiliary region classification branch by adjusting segmentation boundaries however motivated by the ensembling and grouping methods, we proposed MEDOE framework to encourage different experts to learn more balanced distribution in the feature space, and finally adjust classification boundaries. Compared to Region Rebalance, the motivation of our work was completely different and provided different research directions.

**Interpretations of mIoU and mAcc difference:** Region rebalance only explain the cause of mIoU with previous long-tailed methods. Different from Region rebalance, we take a further step and explore the significance of mAcc in segmenting the body and tail categories.

**Datasets difference:** RR used COCO164 (164 categories) and ADE20K (150 categories) datasets, both of which are massive categories datasets. We used Cityscapes (19 categories) and ADE20K to verify both a small number of categories and massive categories scenarios. The decline in mIoU may be partly due to the impact of the dataset, we also demonstrated the gap between Cityscapes and ADE20K in Appendix A.5. Our method demonstrated excellent results on a large number of categories of datasets such as ADE20K.

### A.10   SIGNIFICANCE OF BODY AND TAIL CATEGORIES IN REAL-WORLD SCENARIOS

We believe in a large number of real-world scenarios, it is more important to be able to identify body and tail categories than accurately segment the edge pixels of head categories. (*i.e.*In the automatic driving scenario, we need to segment some tail categories objects that appear on the driving path, such as "poles" or "fire hydrants", to avoid traffic accidents. Segmenting small lesions on medical images can help doctors detect underlying diseases). Generally, the benefits of this segmentation are far greater than the decrease of certain head categories edges.

### A.11   COMPARISON WITH TRADITIONAL MULTI-EXPERTS METHODS

We compared the differences from existing multi-expert methods, there is three main difference:

**Model architecture:** The pipeline of traditional multi-expert methods contain a backbone and multi-classifiers to adjust classifier boundaries and finally ensemble the outputs. 1)We pioneered the **combination of contextual modules and classifiers** to become experts and learn more balanced distribution in the feature space, and finally adjust classification boundaries. 2)Then we provided each expert with soft weight based on the final contextual information and classification results through a learning mechanism to ensemble the outputs.

**Training strategies:** Traditional methods often focus on constraining dominating categories and ignoring the confusing categories. Differing from them, we proposed the expert-specific pixel-

masking strategy and diverse data distribution-aware loss function to ensure our model architecture focuses on the confusing categories and has better performance.

**Training step:** Advanced multi-expert methods in long-tailed classification, such as BBN (Zhou et al., 2020), RIDE (Wang et al., 2020b), LFME (Xiang et al., 2020). They all take multiple steps to train, mainly including 1. training backbone, 2. training classifiers, and 3. Distillation learning (optional). However, our method can update the backbone parameters while training the head expert, so it is one-step training.

