# OpenReview forum: "MEDOE: A Multi-Expert Decoder and Output Ensemble Framework for Long-tailed Semantic Segmentation"
_ICLR.cc/2023/Conference — Submitted to ICLR 2023_

### Official Review · Reviewer_Yf4C · 2022-10-23

**Confidence:** 4
**Clarity, Quality, Novelty And Reproducibility:** Writing is good.
**Correctness:** 3
**Technical Novelty And Significance:** 2
**Empirical Novelty And Significance:** 2
**Recommendation:** 5

**Strength And Weaknesses:**

The paper introduces a hierarchical structure, improves the quality of long-tailed distribution in semantic
segmentation. The experiments show the competitive performance of the proposed method. The paper provides detailed information for reproduction, opens a new direction for future research.

There are some questions.

1.Equation (4) looks strange.

2.In Table 2, the results on Cityscapes is consistently lower than baseline accross different method, I think something should be done to improve the experimental results

**Summary Of The Paper:**

This paper is about semantic image segmentation. The authors notices that there are some categories have less pixels than other categories, and propose to divide all the categories into different groups according to number of pixels and proposes to use mixture models to different group of categories. Experiments on Cityscapes and ADE20k prove the effectiveness.

**Summary Of The Review:**

This paper proposes to use a two-stage model for semantic image segmentation. The idea is straight-forward and the main question is experiment, which is not good enough.

---

> ### Author Response · Authors · 2022-11-18
> **Thanks so much for your detailed questions and constructive comments.**
>
> Thanks so much for your detailed questions and constructive comments. We address each of your comments below.
>
> **[Q1]: Equation (4) looks strange.**
>
> **[A1]**: **The interpretation of Eq.(4).** Eq.(4) is the formula for auxiliary loss.
>
> * The first term is an L2 regularization term to decline the confusion of  $S^{IC}_i$ and make  experts focus on the dominating categories through minimizing all categories logits in $S^{IC}_i$;
> * The second term is the sum of a KL-divergence between the expert classification probability **p** with the reality label distribution **q** in ground truth *y* over $S_i$ categories. We minimize this term to let our experts' classifications closer to ground truth and avoid the over-confidence caused by the first term.
>
> **Revised manuscript:**  We have rewritten the description of auxiliary loss and Eq.(4) in Sec 3.1.
>
>
>
>
>
> **[Q2]: In Tabel 2, the results on Cityscapes is consistently lower than baseline accross different method, I think something should be done to improve the experimental results.**
>
> **[A2]**: Thanks for your positive suggestion. We need to highlight, our model MEDOE mIoU metrics improvement is minor, but mAcc improvement is significant in Cityscapes. As mentioned in Sec 3.3 and Appendix A.5, we demonstrated that mAcc is more important for body and tail categories in long-tailed segmentation and the reason for the improvement in Cityscapes is minor. Specifically:
>
> 1. We have analyzed the performance gap between Cityscapes and ADE20K. (cf. Appendix A.5)  Fewer categories in Cityscapes and a **more pronounced long-tail distribution** (higher proportion of head category instances), so it is easier to fall into local optimum when training body and tail experts.
> 2. We demonstrated the importance of the huge boost in mAcc.
>    * **The huge improvements in mAcc have significance to segment the body and tail categories.** In Cityscapes, mIoU is indeed kept stable, and we also analyzed the reasons for the gap between Cityscapes and ADE20K in Appendix A.5. We conclude that our approach will perform better in massive-category datasets, which also meet real-world scenarios. Meanwhile, we also demonstrated that mAcc is a more tail-sensitive and fairer metric of body and tail categories while keeping mIoU sTabel (cf. Sec 3.3 and Appendix A.2).
>    * **The significance of segmenting body and tail categories in real-world scenarios.** We focus on the segmentation effect of body and tail, and we believe that in a large number of scenarios, it is more important to be able to identify tail categories than to accurately segment the edge pixels of head categories(i.e., some tail categories objects that appear on the driving path in automatic driving, such as ``poles", or some small lesions in medical images). The huge improvement in mAcc shows the effectiveness of our approach on tail and body categories.
>
> **Revised manuscript:**  We have added additional explanations in A.10.

---

### Official Review · Reviewer_Vw5N · 2022-10-23

**Confidence:** 4
**Correctness:** 2
**Technical Novelty And Significance:** 2
**Empirical Novelty And Significance:** 2
**Recommendation:** 5

**Clarity, Quality, Novelty And Reproducibility:**

- Some arguments need fine-tuning to improve the clarity and quality in the paper, especially when it comes to formalizing them. Another example to the ones above is the last paragraph on page 4: X and Y should be apparently defined as a tuple (X, Y), but the definition of Y as a sample-agnostic label set is confusing.
- There is limited novelty in this work. The approach to process groups of categories with a standalone classifier is a common theme in the literature on long-tail recognition.
- The implementation details seem sufficient for reproducibility.

**Strength And Weaknesses:**

Strengths:
- I concur with the authors that the long-tail problem in recognition deserves more research.
- The proposed approach is intuitive in its intention and improves the mean accuracy.
- Experiments with multiple model architectures and datasets.
- The reference numbers for the oracle scenario are interesting.

Weaknesses:

I understand that the approach does not pan out in terms of the IoU, hence the need to push mAcc as the more important metric. The work provides some formal argument, which I do not find convincing, however. The interpretation of the results — mAcc improves, mIoU does not — has a simple explanation to me: the method reduces the number of false negatives at the cost of increasing the number of false positives. I do not consider this is an overall improvement.

Some technical steps are not explained to a sufficient extent:
- The discussion leading up to Eq. 4 is unclear. What norm is imposed on z and why this need? What is the “reality” data distribution?

The notation has many loose ends, which impedes understanding of the actual implementation:
- In Eq. 4 the sum scope is unclear;
- The second line above Eq. 6: the condition for y membership is unclear, because the sets S are disjoint, as defined above.

Some of the claims are not backed up:
- Second line after Eq. 5: “K experts are good and distinctive from each other” — how are the goodness and distinctiveness measured?

**Summary Of The Paper:**

The paper tackles the problem of improving the accuracy of semantic segmentation on long-tail classes. To this end, it introduces separate heads to the network output that focus only on specific segments of the class distribution, which are grouped heuristically into “head”, “body” and “tail” categories, based on their frequency in the dataset. A few decoder layers are trained to aggregate the predictions from these heads into the final model prediction. This approach improves the mean pixel accuracy across classes, but mIoU stays largely the same.

**Summary Of The Review:**

The empirical results do not show a significant improvement in terms of mIoU, which accounts for false positives and false negatives. The argument that mAcc is a more important measure is not convincing, since one would need to consider a very specific scenario of a downstream task for semantic segmentation. Since we commonly study semantic segmentation in an application-agnostic way, mIoU remains an important metric even if the focus is on the long-tail categories.

---

> ### Author Response · Authors · 2022-11-18
> **Thanks so much for your detailed questions and constructive comments. (Part 1/3)**
>
> Thanks so much for your detailed questions and constructive comments. We address each of your comments below.
>
> **[Q1]: I understand that the approach does not pan out in terms of the IoU, hence the need to push mAcc as the more important metric. The work provides some formal argument, which I do not find convincing, however. The interpretation of the results — mAcc improves, mIoU does not — has a simple explanation to me: the method reduces the number of false negatives at the cost of increasing the number of false positives. I do not consider this is an overall improvement.**
>
> **[A1]**: Thanks for your positive comments. We need to highlight, our model MEDOE mIoU metrics improvement is minor, but mAcc improvement is significant. As mentioned in Sec 3.3 and Appendix A.2, we argue that mAcc is more important for body and tail categories in long-tailed segmentation. Specifically:
>
> 1.  The selection of mIoU and mAcc:
>     * **MIoU is a metric of overall (whole image) performance.** As shown in the experimental results we have not abandoned the importance of mIoU as a metric to evaluate the segmentation effect of the whole image.
>     * **MAcc is a metric of a tail-sensitive metric.** The point we made in the selection of mIoU and mAcc evaluation metrics is that mIoU is a more holistic evaluation metric. The $ Acc_i $ formula in mAcc is only related to its categories, while $IoU_i$ is affected by $FP_i$ (cf. Sec 3.3 and Appendix A.2). We proved theoretically that $FP_i$ is dominated by the category of the head, and we believed that IoU of body and tail will be dominated by the category of head and cannot reflect the effect of solving the long-tailed issue. Thus we demonstrated that mAcc is a more tail-sensitive and fairer metric of body and tail categories while keeping mIoU stabel.
> 2.  **We demonstrated the reason for the significance of segment tail and body categories in real-world scenarios:**  We focus on the segmentation effect of body and tail, and we believe that in a large number of scenarios, it is more important to be able to identify tail categories than to accurately segment the edge pixels of head categories(i.e., some tail categories objects that appear on the driving path in automatic driving, such as ``poles", or some small lesions in medical images).
>
> **Revised manuscript:**  We have added the significance of segment tail and body categories in real-world scenarios in Appendix A.10.
>
>
>
>
>
> **[Q2]: Some technical steps are not explained to a sufficient extent:**
>
> **The discussion leading up to Eq.(4) is unclear. What norm is imposed on z and why this need? What is the “reality” data distribution?**
>
> **[A2]**: Thanks for your detailed questions, we will explain the questions below, respectively.
>
> ​		1) **Q:** What norm is imposed on z and why this need?
>
> ​			**A:** For $z$ we adopt an L2 regularization to decline the confusion of $S^{IC}_i$ and make experts focus on the dominating categories through minimize all categories logits in $S^{IC}_i$;
>
> ​		2) **Q:** What is the “reality” data distribution
>
> ​		 	**A:** The reality distribution **'q'** denotes the really label distribution of ground truth $y$ in each expert dominating Categories ${S}_{i}$.
>
> ​		3) **Q:** Confusion of Eq.(4):
>
> ​			 **A:** Eq.(4) is the formula for auxiliary loss.
>
> * The first term is an L2 regularization term to decline the confusion of  $S^{IC}_i$ and make  experts focus on the dominating categories through minimizing all categories logits in $S^{IC}_i$;
> * The second term is the sum of a KL-divergence between the expert classification probability **p** with the reality label distribution **q** in ground truth *y* over $S_i$ categories. We minimize this term to let our experts' classifications closer to ground truth and avoid the over-confidence caused by the first term.
>
> **Revised manuscript:**  We have rewritten the description of auxiliary loss and Eq.(4) in Sec 3.1.

---

> > ### Author Response · Authors · 2022-11-18
> > **Thanks so much for your detailed questions and constructive comments. (Part 2/3)**
> >
> > **[Q3]: The notation has many loose ends, which impedes understanding of the actual implementation:**
> >
> > **[A3]**: Sorry for the misunderstanding of your reading.
> >
> > ​		1) Q: In Eq.(4) the sum scope is unclear;
> >
> > ​		A: In this paper,  $S_i$ and  $S^{IC}_i$ represent the dominating and interfering categories of experts respectively, so the sum scopes are the sum of the L2 norm term of all interfering categories and the KL divergence of all dominant categories, respectively.
> >
> > ​		2) Q: The second line above Eq.(6): the condition for y membership is unclear, because the sets S are disjoint, as defined above.
> >
> > ​		A: **There is overlap between  $S_i$ of each subset divided by our expert-specific pixel-masking strategy,** for the following reasons:
> >
> > 1. **Overlap does not cause a decline in the performance of the majority categories.** The fact that majority categories can dominate the model has been demonstrated, (this is also the essence of the long-tailed distribution). Thus when we kept the tail when segmenting the head categories, the tail categories will not greatly affect the effectiveness of the head expert;
> > 2. **Overlap can help ensembling results.**
> >    * We considered about the selection of Multi-expert output is not entirely correct. (this is why there is a gap between our current results and Oracle results). If we do not set up the overlap, we will be overfitting in the dominating categories, and get a low-level performance;
> >    * Our overlap setting can also ensure that fewer categories are exposed to more experts, so that when ensembling the results, experts in tail categories can also provide auxiliary advice for the dominating experts (this is also the reason for our set of soft weight)
> >
> > Under this setting, there is an intersection between our $S_i$ and $S_{i+1}$, so **$y$ needs to meet the setting of the second line above Eq.(6).**
> >
> > **Revised manuscript:**  We have rewritten the description of auxiliary loss and Eq.(4) in Sec 3.1 , and added the brief reasons for adopting overlap below Eq.(1) (cf. Sec 3.1) .
> >
> >
> >
> > **[Q4]: Some of the claims are not backed up:**
> >
> > Second line after Eq.(5): “K experts are good and distinctive from each other” — how are the goodness and distinctiveness measured?
> >
> > **[A4]**: Thanks for your detailed question.  We added **mAcc and bias of *head, body and tail* between different experts** to measure the goodness and distinctiveness.
> >
> > 1. **The design principle of the loss function.** Before this sentence, we required that each expert performed better on their dominating categories by designing a diverse data distribution-aware loss function, Thus forming "K experts are good and distinctive from each other".
> > 2. **Experimental results for multi-experts.** In addition, to measure the goodness and distinctiveness of each expert, we added experiments about mAcc and bias in different splits between experts in Appendix A.8, and we showed the results below. (The higher mean accuracy and lower bias are better)
> >
> > |          | Many<br />mAcc  bias | Medium<br />mAcc  bias | Few<br />mAcc  bias |
> > | -------- | :------------------: | :--------------------: | :-----------------: |
> > | Overall  |      0.96  0.10      |       0.88  0.22       |     0.81  0.30      |
> > | Expert 1 |      0.97  0.09      |       0.86  0.24       |     0.75  0.36      |
> > | Expert 2 |       -     -        |       0.95  0.09       |     0.86  0.19      |
> > | Expert 3 |       -     -        |        -     -         |     0.91  0.13      |
> >
> > **Revised manuscript:** We have added experiments in Appendix A.8.

---

> > > ### Author Response · Authors · 2022-11-18
> > > **Thanks so much for your detailed questions and constructive comments. (Part 3/3)**
> > >
> > > **[Q5]: Some arguments need fine-tuning to improve the clarity and quality in the paper, especially when it comes to formalizing them. Another example to the ones above is the last paragraph on page 4: X and Y should be apparently defined as a tuple (X, Y), but the definition of Y as a sample-agnostic label set is confusing.**
> > >
> > > **[A5]**: Thanks for your positive comments. We have changed the original {X; Y} to {(X,Y); C}. “Given a training set $D=\{(X,Y);C\}$,$X$ denotes the data and $Y$ denotes ground truth labels, with $C$ categories in total.”
> > >
> > > **Revised manuscript:**We have rewritten the paragraph in Sec 3.1.
> > >
> > >
> > >
> > >
> > >
> > > **[Q6]: Concerns about our work generalizability of sentence ``The argument that mAcc is a more important measure is not convincing, since one would need to consider a very specific scenario of a downstream task for semantic segmentation. Since we commonly study semantic segmentation in an application-agnostic way, mIoU remains an important metric even if the focus is on the long-tail categories." in summary.**
> > >
> > > **[A6]**: Thanks for your consideration of our work. We do not think our work ``need to consider a very specific scenario". As we mentioned, different from ciffar-LT or Imagenet-LT, **semantic segmentation is a natural long-tailed distribution scenario**. Therefore, there is a huge significance to segmenting the categories of body and tail in almost all real-world scenarios. MIoU as an overall metric should indeed be focused on, but the importance of mAcc for body and tail segmentation should also be recognized.
> > >
> > > **Revised manuscript:** We have added the discussion about the significance of body and tail categories segmentation in Appendix A.10.

---

### Official Review · Reviewer_FVTB · 2022-10-25

**Confidence:** 4
**Correctness:** 4
**Technical Novelty And Significance:** 3
**Empirical Novelty And Significance:** 3
**Recommendation:** 5

**Clarity, Quality, Novelty And Reproducibility:**

The clarity, quality and novelty is good. The reproducibility cannot be told since the code is not contained.

**Strength And Weaknesses:**

Strength
The paper is well-written, and the experiments are well built to cover two most popular datasets for semantic segmentation. The ablation is comprehensive to cover the main contribution of this paper

Weakness

1. What is the contextual module? It needs to be illustrated with more details.

2. The auxiliary loss is unclear to me. What is the reality distribution 'q'. I do not quite understand this comment 'It is inevitable that we need to classify these pixels, even if they fall into majority categories'. From my understanding, for experts that process body or tail categories, these majority categories can be ignored without causing any gradient. In equation 4, the auxiliary loss tends to minimize the logits for these classes, which needs more explanation. In summary, the role played by this loss is quite confusing.

3. I have a big concern about the effectiveness of the proposed method. In table 2, the performance of the proposed method is even worse than the baseline. Despite the oracle results are promising, it is not applicable since we cannot priorly know which expert should process which pixel. For table 4, I can see some improvement for the rare classes on cityscapes, but the improvement is also quite minor. I admit the improvement on ADE20k is interesting, but only obtaining good results on one dataset is not enough to validate the effectiveness of the proposed method.

4. The paper divides the categories into three sets. But during training, the experts process some overlapped categories. Why does the author device such an architecture instead of letting each expert work on a separate class set.

5. The author only compare their method to the straight baseline. It is okay but it would be better if it can comprehensively compare the method to the SOTA long-tail techniques used in semantic segmentation.

**Summary Of The Paper:**

The paper considers the semantic segmentation problem. The motivation of this paper is based on the relatively poor performance for the tailed classes and in order to overcome this issue, the author propose to use multiple experts to process pixels with different category frequency. In the experimental section, the paper illustrates some improvements especially for the rare classes on Cityscapes and ADE20k datasets.

**Summary Of The Review:**

Overall speaking, the paper is interesting, but it has a big space to be further improved. I would like to see the response of the author to my questions.

---

> ### Author Response · Authors · 2022-11-18
> **Thanks so much for your detailed questions, and insightful and constructive comments. (Part 1/3)**
>
> Thanks so much for your detailed questions, and insightful and constructive comments. We address each of your comments below.
>
> **[Q1]: What is the contextual module? It needs to be illustrated with more details.**
>
> **[A1]**: Sorry for the brief introduction to the context module!
>
> **Contextual module**, as an important module in semantic segmentation, refers to the extraction and aggregation of contextual information for pixels through a series of operations (\ie, feature pyramids, atrous convolution, large-scale convolutional, attention mechanisms, and global pooling). Existing contextual modules include ASPP [R1], PSP [R2], or non-local [R3].
>
> By ``**contextual information** ’’means the relationship between this pixel and the surrounding pixels and global information.
>
> **The reasons why segmentation needs a contextual module:** When solving semantic segmentation tasks if each pixel considers only its deep features, such as texture and color, it will be difficult to classify into the correct category (such as the deep features of leaves and green grass will be very similar). As stated in the work, semantic segmentation believes that each pixel is not I.I.D.. but is related to the surrounding pixels. The correlation information can better help the segmentation task (such as the leaves are surrounded by branches, and the grass is likely to be surrounded by roads).
>
> **Revised manuscript:** We've added more descriptions in Sec 1 and Appendix A.7 in the paper.
>
> [R1]: Chen et al. Encoder-decoder with atrous separable convolution for semantic image segmentation. ECCV. 2018: 801-818.
>
> [R2]: Zhao H et al. Pyramid scene parsing network. CVPR. 2017: 2881-2890.
>
> [R3]: Wang X et al. Non-local neural network. CVPR. 2018: 7794-7803.
>
>
>
>
>
> **[Q2]:The auxiliary loss is unclear to me. What is the reality distribution 'q'. I do not quite understand this comment 'It is inevitable that we need to classify these pixels, even if they fall into majority categories'. From my understanding, for experts that process body or tail categories, these majority categories can be ignored without causing any gradient. In equation 4, the auxiliary loss tends to minimize the logits for these classes, which needs more explanation. In summary, the role played by this loss is quite confusing.**
>
> **[A2]**: Sorry about the confusion about the auxiliary loss. We detailed clarify all the mentioned confusing concepts below:
>
> ​		1) **Meaning of Reality distribution 'q'**. The reality distribution **'q'** denotes the really label distribution of ground truth $y$ in each expert dominating Categories $S_i$.
>
> ​		2)  **Meaning of the sentence 'It is inevitable that ... majority categories'.** This comment is ambiguous, what we want to express is about interfering categories (IC). Based on the expert-specific pixel-masking strategy, the labels have been divided into different sets for experts. IC means that the pixels categories are not in $S_i$. For example, the head categories are IC for body categories. Since we expect each expert to focus on the dominating categories, $S^{IC}_{i}$is inevitably the main source of confusion $E_i$
>
> ​		3) **The interpretation of Eq.(4).** Eq.(4) is the formula for auxiliary loss.
>
> * The first term is an L2 regularization term to decline the confusion of  $S^{IC}_i$ and make  experts focus on the dominating categories through minimizing all categories logits in $S^{IC}_i$;
> * The second term is the sum of a KL-divergence between the expert classification probability **p** with the reality label distribution **q** in ground truth *y* over $S_i$ categories. We minimize this term to let our experts' classification closer to ground truth and avoid the over-confidence caused by the first term.
>
>  **Revised manuscript:**  We have rewritten the description of auxiliary loss and Eq.(4) in Sec 3.1.

---

> > ### Author Response · Authors · 2022-11-18
> > **Thanks so much for your detailed questions, and insightful and constructive comments. (Part 2/3)**
> >
> > **[Q3]: I have a big concern about the effectiveness of the proposed method. In Tabel 2, the performance of the proposed method is even worse than the baseline. Despite the oracle results are promising, it is not applicable since we cannot priorly know which expert should process which pixel. For Tabel 4, I can see some improvement for the rare classes on cityscapes, but the improvement is also quite minor. I admit the improvement on ADE20K is interesting, but only obtaining good results on one dataset is not enough to validate the effectiveness of the proposed method.**
> >
> > **[A3]**: Thanks for your positive suggestion. We need to highlight, our model MEDOE mIoU metrics improvement is minor, but mAcc improvement is significant. As mentioned in Sec 3.3 and Appendix A.2, we argued that mAcc is more important for body and tail categories in long-tailed segmentation. Specifically:
> >
> > 1. **The huge improvements in mAcc have significance to segment the body and tail categories.** In Cityscapes, mIoU is indeed kept stable, and we also analyzed the reasons for the gap between Cityscapes and ADE20K in Appendix A.5. We concluded that our approach will perform better in massive-category datasets, which also meet real-world scenarios. Meanwhile, we also demonstrated that mAcc is a more tail-sensitive and fairer metric of body and tail categories while keeping mIoU s (cf. Sec 3.3 and Appendix A.2).
> > 2. **In real-world scenarios, segmented body and tail categories have huge significance.** We believe that in a large number of scenarios, it is more important to be able to identify body and tail categories than to accurately segment the edge pixels of head categories (i.e., some tail categories objects that appear on the driving path in automatic driving, such as "poles" and "fire hydrants", or some small lesions in medical images). The huge improvement in mAcc shows the effectiveness of our approach on tail and body categories.
> >
> > **Revised manuscript:** We have added additional explanations in Appendix A.2 and A.10.
> >
> >
> >
> >
> >
> > **[Q4]: The paper divides the categories into three sets. But during training, the experts process some overlapped categories. Why does the author device such an architecture instead of letting each expert work on a separate class set.**
> >
> > **[A4]**: Thanks for your detailed question. In the design of overlap, we mainly considered the following points:
> >
> > 1. **Overlap does not cause a decline in the performance of the majority categories.** The fact that majority categories can dominate the model has been demonstrated, (this is also the essence of the long-tailed distribution), so when we keep the tail when segmenting the head categories, the tail categories will not greatly affect the effectiveness of the head expert;
> > 2. **Overlap can help ensembling results.**
> >    * We considered about the selection of multi-experts outputs is not entirely correct. (this is why there is a gap between our current results and Oracle results). If we do not set up the overlap, we will be overfitting in the dominating categories, and get a low-level performance;
> >    * Our overlap setting can also ensure that fewer categories are exposed to more experts, so that when ensembling the results, experts in tail categories can also provide auxiliary advice for the dominating experts (this is also the reason for our set of soft weight)
> >
> > **Revised manuscript:** We've added the brief reasons for adopting overlap below Eq.(1) (cf. Sec 3.1) in revised manuscript .

---

> > > ### Author Response · Authors · 2022-11-18
> > > **Thanks so much for your detailed questions, and insightful and constructive comments. (Part 3/3)**
> > >
> > > **[Q5]: The author only compare their method to the straight baseline. It is okay but it would be better if it can comprehensively compare the method to the SOTA long-tail techniques used in semantic segmentation.**
> > >
> > > **[A5]**: Thanks for your positive comments. We have done some comparisons about the long-tailed classification methods in segmentation.
> > >
> > > 1. **There is no SOTA long-tail techniques used in semantic segmentation.** To the best of our knowledge, our work is the first to explicitly focus on long-tailed semantic segmentation. RR [R1] can be regarded as a contemporaneous work. Unfortunately, since the code of RR [R1] has not been open source, we can not implement it in the short term, so there’s no way to make a fair comparison.
> > >
> > > 2. **Experiments compared with advanced long-tailed methods in classification/detection.** To better reflect the contribution of our work in long-tailed semantic segmentation, we have taken the MC approach [R2] in Tabel 5 of Sec 4.2 and Resampling [R3], reweighting methods (cf. Appendix. Tabel 8) to demonstrate the applicability of the previous ensembling and grouping, resampling, and reweighting methods in solving the semantic segmentation of long-tail distributions.
> > >
> > >    In addition, we conducted some ablation studies on Cityscapes and ADE20K to further explore this applicability, thus reflecting the contribution of our work, we complemented the experimental results of LDAM and SeasawLoss on semantic segmentation in Appendix Tabel 8.
> > >
> > > | Benchmarks | Methods                |     mIoU      |     mAcc      |
> > > | ---------- | ---------------------- | :-----------: | :-----------: |
> > > | Cityscapes | Baseline               |     80.37     |     85.68     |
> > > |            | Reweighting-FocalLoss  | 76.23(-4.14)  | 85.00(-1.68)  |
> > > |            | Reweighting-LDAMLoss   | 77.52(-2.85)  | 85.15(-1.43)  |
> > > |            | Reweighting-SeesawLoss | 67.58(-12.79) | 74.35(-12.33) |
> > > |            | Re-sampling            | 66.79(-13.58) | 75.21(-11.47) |
> > > |            | Baseline+MED           | 80.20(-0.17)  | 90.04(+3.36)  |
> > > | ADE20K     | Baseline               |     42.11     |     54.13     |
> > > |            | Reweighting-LDAMLoss   | 37.61(-4.50)  | 55.29(+1.16)  |
> > > |            | Reweighting-SeesawLoss | 40.39(-1.72)  | 48.78(-5.35)  |
> > > |            | Re-sampling            | 33.87(-8.24)  | 40.19(-13.94) |
> > > |            | Re-sampling            |       -       |       -       |
> > > |            | Baseline+MED           | 80.20(+1.71)  | 60.02(+5.89)  |
> > >
> > > **Revised manuscript:** We added ablation experiments (SeasawLoss [R5]& LDAMLoss [R4]) on Cityscapes and ADE20K in Appendix Tabel 8 and A.11 to further explore this applicability, thus reflecting the contribution of our work.
> > >
> > > [R1] Region Rebalance for Long-Tailed Semantic Segmentation , In arXiv, 2022.
> > >
> > > [R2] Kang et al. Decoupling representation and classifier for long-tailed classification. ICLR, 2020.
> > >
> > > [R3] Lin T Y et al. Focal loss for dense object detection. ICCV 2017: 2980-2988.
> > >
> > > [R4] Cao K et al. Learning imbalanced datasets with label-distribution-aware margin loss. NIPS. 2019, 32.
> > >
> > > [R5] Wang J et al. Seasaw loss for long-tailed instance segmentation. CVPR. 2021: 9695-9704.
> > >
> > >
> > >
> > >
> > >
> > > **[Q6]: The reproducibility cannot be told since the code is not contained.**
> > >
> > > **[A6]**: Thanks for your positive comments. After our paper is successfully accepted, **we will open source our code as soon as possible.**

---

### Official Review · Reviewer_9fjo · 2022-11-03

**Confidence:** 3
**Correctness:** 2
**Technical Novelty And Significance:** 2
**Empirical Novelty And Significance:** 2
**Recommendation:** 5

**Clarity, Quality, Novelty And Reproducibility:**

This paper is of good clarity and quality in writing, and is well-organized. Yet, it should have compared with some existing methods in long-tailed classification, which also provide semantic segmentation results. Besides, the novelty is rather limited. Such a method is widely used in multi-expert long-tail classification/detection task. As for the results, I believe that such results can be easily reproduced.

**Strength And Weaknesses:**

Strength；
1.	This paper makes a survey on the topic of long-tail problem in semantic segmentation.
2.	It proposes a model-agnostic multi-expert decoder to resolve such a problem and makes certain improvements for some classical segmentation models.
3.	A diverse data distribution-aware loss function is proposed for preventing over-confidence of minority categories.
4.	The paper conducts some analyses to give the ideal results for the proposed method, showing a promising direction of future research.

Weakness;
1.	This paper claims that it is the first to explicitly focus on the long-tailed semantic segmentation. Yet, There are already some published papers which concentrate on such topic (eg, Region Rebalance for Long-Tailed Semantic Segmentation ,cvpr2022)
2.	The paper duplicate the decoder head twice, bringing a large quantity of extra params. I am not pretty sure that whether the improvements are resulted from such a change.
3.  The overall pipeline does not differ much from some multi-expert methods in long-tail classification/detection. I would recommend to compare with some of them, as they also conduct segmentation experiments such as "Distribution Alignment: A Unified Framework for Long-tail Visual Recognition"
4. The paper propose that the mAcc metric is a more important metric for long-tail segmentation, giving degenerated mIoU results and better mAcc results in some experiments. Yet, this problem is also mentioned by paper "Region Rebalance for Long-Tailed Semantic Segmentation in cvpr2022" and has been well addressed.



**Summary Of The Paper:**

This paper focuses on the topic of long tail in semantic segmentation. It designs a model-agnostic multi-expert decoder and output framework, making certain improvements for some classical segmentation models. A diverse data distribution-aware loss function is proposed for preventing over-confidence of minority categories. Besides, it advocates mAcc as a more important metric to evaluate the performance for body and tail categories in long-tailed semantic segmentation. The paper demonstrates some analyses to prove such opinion.

**Summary Of The Review:**

This paper presents the long-tail problem in semantic segmentation and claims that they are the first to explicitly focus on the long-tailed semantic segmentation. Yet, there are already some published papers focusing on such topic (eg, Region Rebalance for Long-Tailed Semantic Segmentation, cvpr2022). Besides, the main idea of this paper is widely adapted by multi-expert long-tailed methods. The paper also advocates that the mAcc metric is more important for long-tail segmentation, and gives degenerated mIoU results. Yet, the paper, Region Rebalance for Long-Tailed Semantic Segmentation, also encounters such a problem and well address this problem.

---

> ### Author Response · Authors · 2022-11-18
> **Thanks so much for your detailed questions and constructive comments. (Part 1/3)**
>
> Thanks so much for your detailed questions and constructive comments. We address each of your comments below.
>
> **[Q1]: Comparison with existing work RR [R1]**
>
> \```
>
> <font color = gray>This paper claims that it is the first to explicitly focus on the long-tailed semantic segmentation. Yet, There are already some published papers which concetopicse on such topic (eg, Region Rebalance for Long-Tailed Semantic Segmentation ,cvpr2022). Yet, this problem is also mentioned by paper "Region Rebalance for Long-Tailed Semantic Segmentation in cvpr2022" and has been well addressed. </font>
>
> \```
>
> **[A1]**: Thanks for your comments. Compared to existing work RR[R1], ***we have four main differences:***
>
> 1. **Perspective difference**: Although we are all explicitly focused on the long-tailed distribution of semantic segmentation. RR was concerned about solving the problem of category rebalance, while our work was more focused on improving the recognition of tail and body categories and placed special emphasis on the significance of segmenting the body, and tail categories.
>
> 2. **Methods difference:** RR relieved the categories imbalance issue with an auxiliary region classification branch (inspired by Re-balancing in long-tailed) by adjusting segmentation boundaries; our work is motivated by the ensembling and grouping methods, we propose the MEDOE framework to encourage different experts to learn more balanced distribution in the feature space, and finally adjust classification boundaries.
>
>    The motivation of our work is completely different and provides **different research directions**. The results of our Oracle experiment have opened up a very huge prospect and demonstrated the superiority of multi-experts.
>
> 3. **Interpretations of mIoU and mAcc difference:** RR only explains the cause of mIoU with previous long-tailed methods. Different from RR, our work takes a further step and explores the mAcc as a tail-sensitive metric, and increasing mAcc has the huge significance of segmenting the body and tail categories.
>
> 4. **Datasets difference:** RR used COCO164 (164 categories) and ADE20K (150 categories) datasets, both of which are massive categories datasets. We used Cityscapes (19 categories) and ADE20K to verify both a small number of categories and massive categories scenarios. The decline in mIoU may be partly due to the impact of the dataset, we also demonstrated the gap between Cityscapes and ADE20K in Appendix A.5. Our method demonstrated excellent results on a large number of categories of datasets such as ADE20K.
>
> In addition, to the best of our knowledge **RR [R1] has not been published in CVPR2022, but was just published on arXiv in 2022.4.** Our paper can be regarded as a contemporaneous work with RR, thus we think that this paper is the first to explicitly focus on long-tailed semantic segmentation. We and Region Rebalance almost simultaneously focus on the long-tailed data distribution as an important reason for constraining semantic segmentation performance, which proves the significance of solving this issue.
>
> Finally, since the code of RR [R1] is not open source, we can not implement it in the short term, so there’s no way to make a fair comparison.
>
> **Revised manuscript:** We have added a discussion in Appendix A.9.
>
> [R1] Region Rebalance for Long-Tailed Semantic Segmentation , In arXiv, 2022.
>
>
>
> **[Q2]: The paper duplicates the decoder head twice, bringing a large quantity of extra parameters. I am not pretty sure that whether the improvements are resulted from such a change.**
>
> **[A2]**: Thanks for such a detailed question.
>
> We do not consider the improvements are primarily due to the extra parameters, for two reasons:
>
> 1. **The number of parameters in our methods is acceptable.** We have shown the number of parameters of our methods in Tabel 2 and 3. The parameters of our MEDOE on PSPNet with ResNet50 and ResNet101 only increased about **0.1 times** with nearly **1.30%, 0.98% increase in mIoU and 2.85%, 5.19%** in mAcc on ADE20K. . At most, parameters were increased by up to about 0.4 times compared to the baseline.
> 2. **The task is really difficult, so simply increasing parameters can not always make sense.** For example, in Tabel 5, we compared the ensemble model method, which increased nearly **2.0 times** with baseline but the performance increased only **0.12% and 0.33% in mIoU and mAcc** without our training strategy, and we can see that simply increasing the number of parameters does not improve model performance, especially for the tail categories.
>
> **Revised manuscript:** We have highlighted the discussion in Sec 4.2 (cf. Ablation of MED).

---

> > ### Author Response · Authors · 2022-11-18
> > **Thanks so much for your detailed questions and constructive comments.(Part 2/3)**
> >
> > **[Q3]: Compared the overall pipeline with multi-expert methods in long-tail classification.**
> >
> > **\[A3]**: Thanks for your positive comments. We compared the differences from existing multi-expert methods, there are three main differences
> >
> > 1. **Model architecture:** The pipeline of traditional multi-expert methods contains a backbone and multi-classifiers to adjust classifier boundaries and finally ensemble the outputs. 1)We pioneered the **combination of contextual modules and classifiers** to become experts and learn more balanced distribution in the feature space, and finally adjusted classification boundaries. 2)Then we provided each expert with soft weight based on the final contextual information and classification results through a **learning mechanism** to ensemble the outputs.
> > 2. **Training strategies:** Traditional methods often focus on constraining dominating categories and ignoring the confusing categories. Differing from them, we proposed an expert-specific pixel-masking strategy and diverse data distribution-aware loss function to ensure our model architecture focuses on the confusing categories and has better performance (cf. Figure 5 in Sec 4.2).
> > 3. **Training step:** Advanced multi-expert methods in long-tailed classification/detection, such as BBN [R2], RIDE [R3], and LFME [R4]. They all take multiple steps to train, mainly including 1. training backbone, 2. training classifiers, and 3. Distillation learning (optional). However, our method can update the backbone parameters while training the head expert, so it is one-step training.
> >
> > We have done some **ablation experiments** on Cityscapes and ADE20K to further explore the contribution of our work. We have taken the MC approach [R5] in Tabel 5 of Sec 4.2 and the Resampling and reweighting [R6] methods in Tabel 8 of the Appendix A.4 to demonstrate the applicability of the previous ensembling and grouping, resampling, and reweighting methods in solving the semantic segmentation of long-tail distributions, respectively.
> >
> > | Benchmarks | Methods                |     mIoU      |     mAcc      |
> > | ---------- | ---------------------- | :-----------: | :-----------: |
> > | Cityscapes | Baseline               |     80.37     |     85.68     |
> > |            | Reweighting-FocalLoss  | 76.23(-4.14)  | 85.00(-1.68)  |
> > |            | Reweighting-LDAMLoss   | 77.52(-2.85)  | 85.15(-1.43)  |
> > |            | Reweighting-SeesawLoss | 67.58(-12.79) | 74.35(-12.33) |
> > |            | Re-sampling            | 66.79(-13.58) | 75.21(-11.47) |
> > |            | Baseline+MED           | 80.20(-0.17)  | 90.04(+3.36)  |
> > | ADE20K     | Baseline               |     42.11     |     54.13     |
> > |            | Reweighting-LDAMLoss   | 37.61(-4.50)  | 55.29(+1.16)  |
> > |            | Reweighting-SeesawLoss | 40.39(-1.72)  | 48.78(-5.35)  |
> > |            | Re-sampling            | 33.87(-8.24)  | 40.19(-13.94) |
> > |            | Re-sampling            |       -       |       -       |
> > |            | Baseline+MED           | 80.20(+1.71)  | 60.02(+5.89)  |
> >
> > **Revised manuscript:** We added ablation experiments (SeasawLoss [R8] & LDAMLoss [R7]) on Cityscapes and ADE20K in Appendix Tabel 8 and A.11 to further explore this applicability, thus reflecting the contribution of our work.
> >
> > [R2] Zhou B et al. Bbn: Bilateral-branch network with cumulative learning for long-tailed visual recognition. CVPR. 2020: 9719-9728.
> >
> > [R3] Wang X et al. RIDE: Long-tailed Recognition by Routing Diverse Distribution-Aware Experts. ICLR. 2021
> >
> > [R4] Xiang L et al. Learning from multiple experts: Self-paced knowledge distillation for long-tailed classification. ECCV 2020: 247-263.
> >
> > [R5] Kang B et al. Decoupling representation and classifier for long-tailed recognition. ICLR 2020
> >
> > [R6] Lin T Y et al. Focal loss for dense object detection. ICCV 2017: 2980-2988.
> >
> > [R7] Cao K et al. Learning imbalanced datasets with label-distribution-aware margin loss. NIPS. 2019, 32.
> >
> > [R8] Wang J et al. Seesaw loss for long-tailed instance segmentation. CVPR. 2021: 9695-9704.

---

> > > ### Author Response · Authors · 2022-11-18
> > > **Thanks so much for your detailed questions and constructive comments. (Part 3/3)**
> > >
> > > **[Q4]: The paper propose that the mAcc metric is a more important metric for long-tail segmentation, giving degenerated mIoU results and better mAcc results in some experiments.**
> > >
> > > **[A4]:** Thanks so much for your question. We need to highlight, our model MEDOE mIoU metrics improvement is minor, but mAcc improvement is significant. As mentioned in Sec 3.3 and Appendix A.2, we argue that mAcc is more important for body and tail categories in long-tailed segmentation. Specifically:
> > >
> > > 1. **Differ from RR the problem of mIoU and mAcc:** RR only explains the cause of mIoU with previous long-tailed methods. (misalignment between the CELoss and target evaluation metric IoU inspires us to explore what would happen when rebalancing in semantic segmentation. Different from RR, our work take a further step and explores the significance of mAcc in segmenting the body and tail categories.
> > > 2. **We proposed mAcc is a tail-sensitive metric and important for body and tail categories:** As we state in the paper, we focus on the segmentation of body and tail categories. mAcc measures the performance of a certain category itself, the large increase in mAcc reflects the improvement of body and tail categories;
> > > 3. **We demonstrated the reason for why the significance of segment tail and body categories:** Being able to better identify the categories of body and tail is important in semantic segmentation in a large number of scenarios. (i.e., In the automatic driving scenario, we need to segment some tail categories objects that appear on the driving path, such as  "poles" or  "fire hydrants", to avoid traffic accidents. Segmenting small lesions on medical images can help doctors detect underlying diseases). Generally, the benefits of these segmentations are far greater than the decrease of certain head categories edges.
> > >
> > > **Revised manuscript:** We have added discussion in Appendix A.9, A.2 and A.10.

---

### Author Response · Authors · 2022-11-18
**We thank the reviewers for their insightful and overall positive comments. We briefly explain the main concerns of the reviewers here.**

We sincerely thank all reviewers for their time and positive, constructive and encouraging comments regarding the manuscript’s structure, and the method's novelty, clarity and reproducibility. We have revised the new manuscript given all reviewers' comments. In order to facilitate reviewers, we have organized the reviewers' main concerns, and have provided a summary and brief explanation in this section. The main comments are below:

**[Q1]**: **Comparison with existing long-tailed semantic segmentation.**

**[A1]**: To the best of our knowledge, our work is the first to explicitly focus on long-tailed semantic segmentation. Only RR[R1] can be considered as a contemporaneous work, but since its code is not open source, unfortunately, we cannot make a fair comparison with it for a short time.

Meanwhile, to prove the contribution and effectiveness of our work, we have done some **ablation experiments with traditional long-tailed methods** on Cityscapes and ADE20K (cf. Table 5 and Table 8). Especially, we compared **the differences between long-tailed classification methods and our work in detail** and explained why our network can be effective in long-tailed semantic segmentation (cf. Appendix A.11).

**Revised manuscript:** We have added the comparison with existing long-tailed methods in Appendix A.9, A.11 and A.4, respectively.




**[Q2]**: **The metrics' selection between mIoU and mAcc.**

**[A2]**: In the long-tailed semantic segmentation task, both mIoU and mAcc are important metrics. MIoU is more focused on the overall image segmentation effect and **mAcc is a fairer and tail-sensitive metric**. Meanwhile, we demonstrated through theoretical analysis that **huge improvements in mAcc have significance to segment the body and tail categories.** (cf. Sec 3.3 and Appendix A.2)

Furthermore, we explained that the semantic segmentation task is a natural long-tail distribution scenario while **improving the segmentation of body and tail categories has a huge significance in most realistic scenarios.** (cf. Sec 1. and Appendix A.10)

**Revised manuscript:** We have added the explanation about the significance of body and tail categories in real-world scenarios in Appendix A.10.



**[Q3]**: **The unconvincing performance in Cityscapes.**

**[A3]**: We need to highlight, **our model MEDOE mIoU metrics improvement is minor, but mAcc improvement is significant.** As mentioned in **Answer 2 [A2]**, we argued that mAcc is more important for body and tail categories in long-tailed segmentation and explained the reason.



**[Q4]**: **The confusing description of the auxiliary loss （Eq.(4)）**

**[A4]**: As we mentioned (cf. Sec 3.1), we required each expert to perform well in their dominating categories (expert-specific masked dataset). Therefore, **we need to suppress the influence of $S^{IC}_i$ using auxiliary loss**, which includes:

* The first term is an L2 regularization term to decline the confusion of  $S^{IC}_i$ and make  experts focus on the dominating categories through minimizing all categories logits in $S^{IC}_i$;
* The second term is the sum of a KL-divergence between the expert classification probability **p** with the reality label distribution **q** in ground truth *y* over $S_i$ categories. We minimize this term to let our experts' classifications closer to ground truth and avoid the over-confidence caused by the first term.

**Revised manuscript:**  We have rewritten the description of auxiliary loss and Eq.(4) in Sec 3.1.

These are our summary and brief responses to the reviewers' general concerns and suggestions. For more detailed questions, we will respond to the reviewers individually. **Once again, we sincerely thank all reviewers for their time and constructive comments.**

[R1] Region Rebalance for Long-Tailed Semantic Segmentation , In arXiv, 2022.

---

### Author Response · Authors · 2022-12-04
**Once again, we sincerely thank all reviewers for your time and constructive comments.**

Once again, we sincerely thank all reviewers for your time and constructive comments. Hope our explanation and experiments are able to address your inquiries. **Please don't hesitate to reply if you have any further concerns. We will do our best to respond to all your concerns.**

---

### Decision · Program_Chairs · 2023-01-20

**Decision:**

Reject

**Justification For Why Not Higher Score:**

I consider the weaknesses of the paper with respect to experimental results and metric, as well as methodological approach and novelty, too significant to justify acceptance of the paper.


**Justification For Why Not Lower Score:**

N/A

**Metareview: Summary, Strengths And Weaknesses:**

All reviewers for this paper concurred that while the paper addresses a useful problem, there are a number of weaknesses that keep the paper below the acceptance threshold for ICLR. Multiple reviewers expressed similar concerns about the experimental results and the effectiveness of the proposed method, and in particular also questioned the paper’s focus on the mAcc metric. There was also some question about the methodical approach and novelty compared to previous works. After reviewing the paper and author and reviewer comments, I agree with the shared assessment of the reviewers and do not recommend acceptance of this paper at this time.